# V-ATPase V0a1 promotes Weibel–Palade body biogenesis through the regulation of membrane fission

Yasuo Yamazaki*, Yuka Eura, Koichi Kokame*

Department of Molecular Pathogenesis, National Cerebral and Cardiovascular Center, Osaka, Japan

**Abstract** Membrane fission, the division of a membrane-bound structure into two discrete compartments, is essential for diverse cellular events, such as endocytosis and vesicle/granule biogenesis; however, the process remains unclear. The hemostatic protein von Willebrand factor is produced in vascular endothelial cells and packaged into specialized secretory granules, Weibel–Palade bodies (WPBs) at the *trans*-Golgi network (TGN). Here, we reported that V0a1, a V-ATPase component, is required for the membrane fission of WPBs. We identified two V0a isoforms in distinct populations of WPBs in cultured endothelial cells, V0a1 and V0a2, on mature and nascent WPBs, respectively. Although WPB buds were formed, WPBs could not separate from the TGN in the absence of V0a1. Screening using dominant–negative forms of known membrane fission regulators revealed protein kinase D (PKD) as an essential factor in biogenesis of WPBs. Further, we showed that the induction of wild-type PKDs in V0a1-depleted cells does not support the segregation of WPBs from the TGN; suggesting a primary role of V0a1 in the membrane fission of WPBs. The identification of V0a1 as a new membrane fission regulator should facilitate the understanding of molecular events that enable membrane fission.

*For correspondence:
yasuo.yamazaki@ncvc.go.jp (YY);
kame@ncvc.go.jp (KK)

**Competing interest:** The authors declare that no competing interests exist.

## Editor's evaluation

This study is of particular interest to endothelial cell biologists and cell biologists working on intracellular transport. The experiments provide new insights into the contribution of a proton pump and protein kinase D in the formation of Weibel-Palade bodies that contain von Willebrand factor in vascular endothelial cells.

## Introduction

Approximately one-third of newly synthesized proteins, including secretory proteins, transition from the endoplasmic reticulum to the Golgi apparatus, thereby allowing versatile protein modifications, such as disulfide bond formation (*Chen et al., 2005*; *Farquhar and Palade, 1981*; *Klumperman, 2011*). Subsequently, the proteins reach the membrane compartment, called the *trans*-Golgi network (TGN) (*Klumperman, 2011*). At the TGN, the proteins are sorted and packaged as cargos into distinct transport carriers that are targeted to individual subcellular destinations, such as the plasma membrane, endosomes, and/or secretory granules in specialized cells (*De Matteis and Luini, 2008*; *Kienzle and von Blume, 2014*; *Pakdel and von Blume, 2018*; *Anitei and Hoflack, 2011*). Biogenesis of such membrane-bound structures is initiated by the deformation of the TGN membrane, which functions as a donor membrane, to form a bud where cargo proteins are packaged. The formation of nascent buds is sometimes associated with the recruitment of the coat proteins, clathrin, and coat protein I (*Kümmel and Reinisch, 2011*; *Antonny, 2006*). Lastly, the buds separate from the TGN

membrane for transport. The process of division of one membrane into two is termed membrane fission, which is expected to occur in leak-free luminal contents (*Campelo and Malhotra, 2012*; *Renard et al., 2018*). To achieve such a high-fidelity system, cells must employ elaborate machinery; however, the mechanism of membrane fission remains elusive. One plausible reason is the difficulty in segregating the membrane fission process from the other steps of biogenesis in cells due to the small membrane-bound structures, such as secretory vesicles that are 50–100 nm in diameter, formed at the TGN (*Campelo and Malhotra, 2012*; *Renard et al., 2018*).

The von Willebrand factor (VWF) is an extra-large multimeric glycoprotein involved in primary hemostasis by adhering platelets to the sites of vascular injury (*Sadler, 2005*; *Sadler, 1998*; *Wagner, 1990*). The multimers found in plasma vary in size; the largest multimer has a molecular mass of 20 MDa (*Fowler et al., 1985*; *Zheng, 2015*). Larger multimers recruit platelets more efficiently; thus, the multimerization of VWF is crucial for hemostasis (*Federici et al., 1989*). VWF is synthesized in vascular endothelial cells as a single polypeptide with a molecular mass of 350 kDa in the endoplasmic reticulum and immediately forms a disulfide bond-dependent dimer. The dimers are then transported through the Golgi apparatus, where they convert dimers to multimers via disulfide bond formation. The multimers are stored in large secretory granules called Weibel–Palade bodies (WPBs) (*McCormack et al., 2017*; *Metcalf et al., 2008*; *Mourik and Eikenboom, 2017*; *Valentijn et al., 2011*; *Weibel, 2012*). WPBs are endothelial cell-specific, elongated, rod-shaped secretory organelles varying in size from 0.5 to 5 µm. VWF multimers in WPBs are exocytosed in response to several stimulants, such as histamine. In electron tomography, VWF multimers in the WPB lumen are found as striated, paracrystalline matrices called VWF tubules (*Berriman et al., 2009*; *Valentijn et al., 2008*; *Zenner et al., 2007*). As VWF tubules are aligned in parallel along the longitudinal axes of WPBs, it is thought that the formation of VWF tubules drives the formation of the unique, elongated shape of the organelles. In vitro reconstitution studies suggest that VWF tubule formation is a highly coordinated process of the alignment of multiple VWF dimers into a right-handed coil and the conversion of the aligned dimers into large multimers (*McCormack et al., 2017*; *Huang et al., 2008*; *Mayadas and Wagner, 1989*; *Zhou et al., 2011*). This tubulation allows approximately 50-fold compaction of the VWF in WPBs. The tubulation process requires acidic pH and $Ca^{2+}$ in vitro (*Huang et al., 2008*; *Zhou et al., 2011*; *Gerke, 2011*; *Springer, 2011*) thus, VWF is expected to be exposed in such a subcellular environment during maturation, presumably at the TGN and/or in the newly forming WPB buds from the TGN (*Gerke, 2011*; *Springer, 2011*; *Wagner et al., 1986*). The biological significance of the acidic milieu in mature WPBs is also evident. Forced neutralization of the WPB lumen by a chemical treatment causes a loss of visible VWF tubules, resulting in rounding of WPBs (*Michaux et al., 2006*). Neutralization does not reduce agonist-evoked VWF exocytosis; instead, it markedly reduces platelet recruitment because the multimers are unable to disperse efficiently upon exposure to an extracellular neutral environment. Thus, this evidence indicates the biological importance of the subcellular acidic milieu in primary hemostasis mediated by VWF; nevertheless, the mechanism by which endothelial cells achieve such an environment has not yet been addressed.

Vacuolar $H^+$-ATPase (V-ATPase), an ATP-driven proton pump, is a major regulator of the subcellular acidic pH in cells (*Collins and Forgac, 2020*; *Maxson and Grinstein, 2014*; *Nakanishi-Matsui et al., 2010*; *Vasanthakumar and Rubinstein, 2020*). It is a multi-subunit complex composed of two large domains, V0 and V1. V0 is a membrane-embedded domain that translocates protons across the membrane, whereas the V1 domain hydrolyzes ATP. In mammalian cells, several subunits exist in multiple isoforms encoded by distinct genes, suggesting a large structural diversity of the V-ATPase complex in cells. This study focused on identifying the localization of V0a isoforms ('subunit a' in the V0 domain) in WPBs. We further analyzed the separation of newly forming WPB buds from the TGN, and investigated the role of V0a1 in PKD-dependent biogenesis of WPBs.

## Results

### Human umbilical vein endothelial cell culture

Although VWF is the most commonly used vascular endothelial cell marker in vivo (*Wagner, 1990*), not all cultured endothelial cells express VWF, particularly when cells are maintained in rather sparse conditions. To use cultured endothelial cells in physiologically relevant conditions, we optimized our culture and gene transfer conditions. Primary cultured human umbilical vein endothelial cells (HUVECs)

form a stable monolayer with a characteristic cobblestone-like appearance when cells are maintained on fibronectin for 5–6 days. Under these conditions, essentially all cells expressed VWF at a comparable level, and the cellular junction proteins were located along the cell boundary, as expected in differentiated endothelial cells (*Figure 1—figure supplement 1A*). Although our tested lipofection- and electroporation-based transfection considerably disturbed the cellular junction, recombinant lentivirus-mediated expression of EGFP showed virtually no effect (*Figure 1—figure supplement 1B*). Therefore, we used a recombinant lentivirus system throughout this study.

## Cellular V-ATPase activity promotes VWF multimer dispersion

The acidic milieu of the WPB lumen is crucial for the hemostatic function of VWF (*Huang et al., 2008*; *Zhou et al., 2011*; *Gerke, 2011*; *Springer, 2011*; *Michaux et al., 2006*). V-ATPase is a major regulator of the subcellular acidic pH in cells (*Collins and Forgac, 2020*; *Maxson and Grinstein, 2014*; *Nakanishi-Matsui et al., 2010*; *Vasanthakumar and Rubinstein, 2020*). To address whether V-ATPase is a determinant of the acidic milieu of WPBs, we used bafilomycin A1 and concanamycin A, which are well-characterized specific V-ATPase inhibitors (*Collins and Forgac, 2020*). Four hours of exposure to the inhibitors caused rounding of WPBs without affecting the secreted amount of VWF (*Figure 1—figure supplement 2A,B*), as previously observed in chemically neutralized cells (*Michaux et al., 2006*), suggesting the possible role of V-ATPase in WPBs. Note that prolonged exposure to bafilomycin A1 (16 h) is reported to reduce the basal/constitutive secretion of VWF (*Torisu et al., 2013*). Upon exocytosis, VWF tubules in the acidic pH inside WPBs are instantly exposed to the extracellular neutral pH. The rapid change in pH causes the unfurling of VWF tubules, resulting in an efficient dispersion of VWF multimers on the cellular surface (as observed in the histamine-exposed cells in *Figure 1—figure supplement 2C*; *Michaux et al., 2006*). While pretreatment with the V-ATPase inhibitors did not alter the regulated exocytosis induced by histamine (*Figure 1—figure supplement 2D*), it impaired the dispersion of VWF multimers, as observed in the cells exposed to $NH_4Cl$, which neutralizes subcellular acidic compartments. The strings formed by tangled VWF multimers upon histamine exposure of the V-ATPase inhibitor- or $NH_4Cl$-pretreated cells were rather thin and fragile compared to the prominent, thick structures observed in the non-pretreated cells (*Figure 1—figure supplement 2C*). These observations suggest the contribution of V-ATPase activity in the acidic milieu inside WPBs. Indeed, we confirmed the WPB localization of V0c and V1A, two common subunits of the mammalian V-ATPase complex, in HUVECs (*Figure 1—figure supplement 3A,B*).

## The V0a isoforms are found on the distinct populations of WPBs

To gain further insight into the role of V-ATPase, we used confocal microscopy to observe and screen the subcellular localization of multiple V-ATPase subunits by expressing them as EGFP-fusion proteins in HUVECs. As a result, we found that two out of four isoforms of V0a ('subunit a' in the V0 domain) were localized in distinct populations of WPBs. As observed in the EGFP-tagged V0a1 (*Figure 1—figure supplement 4A*), the endogenous V0a1 (isoform 1 of the V0a subunit) was mainly found in WPBs (*Figure 1A*). The signal was evident on the WPBs observed in the cell periphery, whereas virtually no signal was observed on the WPBs in the perinucleus (*Figure 1A*). WPBs are formed at the TGN and transported to the cell periphery upon maturation, suggesting V0a1 localization in mature WPBs. Rab27a, a small GTPase, is a well-characterized molecule recruited to WPBs during maturation (*Bierings et al., 2012*; *Nightingale et al., 2009*; *Rojo Pulido et al., 2011*). Post-budded WPBs from the TGN move along microtubules to the cell periphery, where they are anchored to actin fibers. Rab27a is required for anchoring WPBs to actin. We found well-matched colocalization of V0a1 and EGFP-Rab27a (*Figure 1B*). Thus, V0a1 appears to be localized in mature WPBs in HUVECs.

V0a2 showed the opposite localization to V0a1; V0a2-EGFP was localized only on the perinuclear WPBs but not on the peripheral WPBs (*Figure 2A*). Accordingly, endogenous V0a2 protein was detected in the buds emerging from the TGN as well as on the TGN (*Figure 1—figure supplement 4B*). Super resolution microscopy revealed that V0a2-EGFP is located on the perinuclear WPB buds; however, they are not often found in post-budded WPBs (*Figure 2B*). The signal intensity of VWF found in the V0a2-positive WPBs was rather faint compared to the strong signal observed in the post-budded WPBs (*Figure 2B*). Live cell imaging confirmed that V0a2-EGFP was considerably localized in the newly forming bud and disappeared as soon as the bud segregated from the Golgi apparatus (*Figure 2C* and *Video 1*). The V0a2 buds found around the TGN were completely absent from

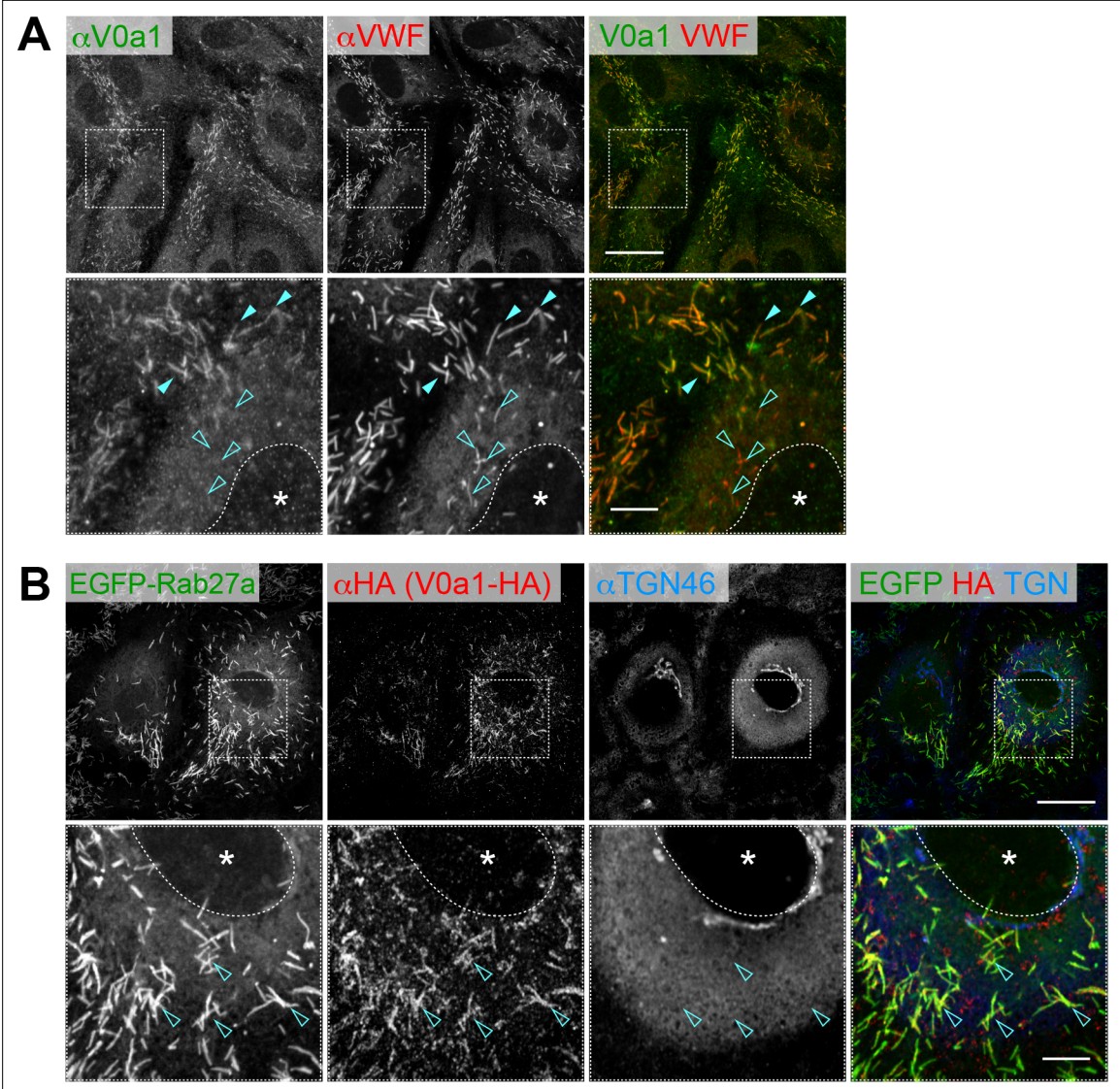

**Figure 1.** V0a1 is localized on peripheral mature Weibel–Palade bodies (WPBs) in human umbilical vein endothelial cells (HUVECs). (**A**) Endogenous V0a1 protein was detected on the peripheral WPBs (closed arrowheads); however, it was not detected on the perinuclear WPBs (open arrowheads). (**B**) V0a1 was well-colocalized with EGFP-Rab27a, a mature WPB marker. Scale bars, 20 μm (wide field) and 5 μm (expanded). Asterisks, nucleus.

The online version of this article includes the following source data and figure supplement(s) for figure 1:

**Figure supplement 1.** Lentivirus-mediated transduction does not show any visible alterations in human umbilical vein endothelial cells (HUVECs).

**Figure supplement 2.** V-ATPase blockage and acidic compartment neutralization cause rounding of Weibel–Palade bodies (WPBs) and impair the efficient dispersion of von Willebrand factor (VWF) multimers.

**Figure supplement 2—source data 1.** Uncropped immunoblot images for *Figure 1—figure supplement 2B and D*.

**Figure supplement 3.** Two common V-ATPase subunits are detected in WPBs in HUVECs.

**Figure supplement 4.** Two V0a isoforms are localized in distinct populations of Weibel–Palade bodies (WPBs) in human umbilical vein endothelial cells.

**Figure supplement 5.** Two AP-1 components, AP1M1-EGFP (**A**) and AP1G1-EGFP (**B**), are localized around the perinuclear WPBs (arrowheads).

**Figure supplement 6.** Subcellular localization of V0a3-EGFP (**A**) and V0a4-EGFP (**B**).

EGFP-Rab27a, a mature WPB marker (*Figure 2D*). These observations indicated the limited localization of V0a2 in nascent WPBs. Newly forming WPB buds in the TGN are clathrin-coated *Zenner et al., 2007*. Depletion of the Golgi-associated clathrin adaptor protein-1 (AP-1) results in an almost total loss of visible WPBs *Lui-Roberts et al., 2005*. Thus, clathrin appears to be an essential factor for biogenesis of WPBs, presumably required for a very early step, such as the bud-forming process at the TGN.

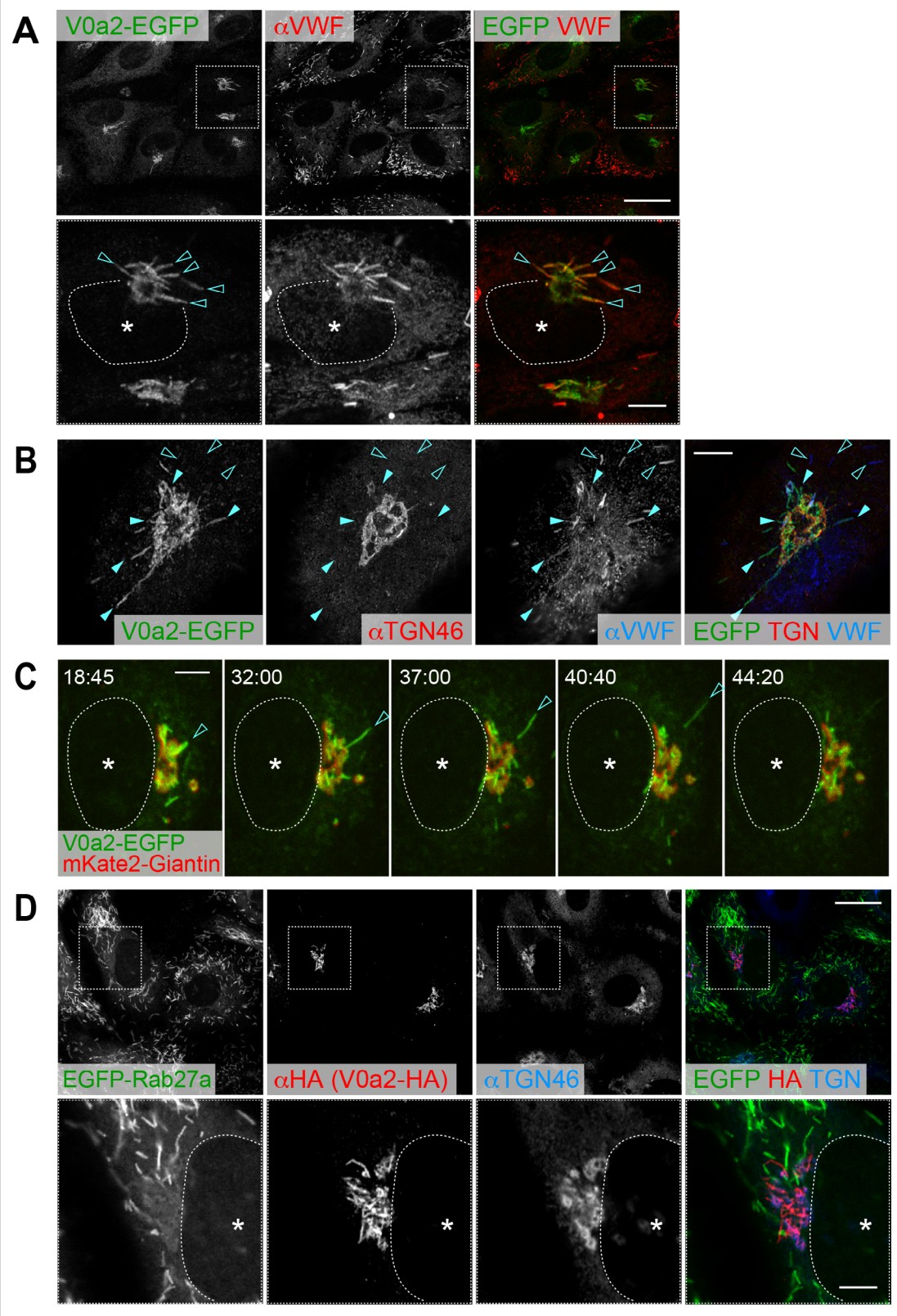

**Figure 2.** V0a2 is localized on the perinuclear newly forming Weibel–Palade bodies (WPBs). (**A**) V0a2-EGFP was found on perinuclear WPBs (arrowheads); however, it was not found in the peripheral WPBs. N-terminal EGFP-tagged V0a2 (EGFP-V0a2) exhibited the same subcellular localization. Scale bars: 20 µm (wide field) and 5 µm (expanded). (**B**) Super resolution image around *trans*-Golgi network (TGN). V0a2-EGFP was observed on buds originating from the TGN (closed arrowheads). The von Willebrand factor signal observed in the V0a2 buds was faint compared to the strong signal

*Figure 2 continued on next page*

*Figure 2 continued*

in the post-budded WPBs (open arrowheads). Scale bar, 5 µm. (**C**) V0a2-EGFP (green) was localized on the newly forming buds in the Golgi apparatus (arrowhead) and disappeared as soon as the bud separates from the Golgi apparatus (mKate2-Giantin, red). Images were captured from the *Video 1*. Scale bar, 5 µm. (**D**) V0a2 buds were absent in Rab27a, a mature WPB marker. Scale bars: 20 µm (wide field) and 5 µm (expanded). Asterisks, nucleus.

The online version of this article includes the following source data and figure supplement(s) for figure 2:

**Figure supplement 1.** Cellular V-ATPase activity is required for von Willebrand factor (VWF) multimerization.

**Figure supplement 1—source data 1.** Uncropped immunoblot images for *Figure 2—figure supplement 1A and B*.

Corroborating the findings of earlier studies, nascent WPBs found in the perinucleus were partially decorated with the components of AP-1 complex (*Figure 1—figure supplement 5A,B*); however, this was not as evident as observed in V0a2-EGFP (*Figure 2A–B*). As the two other V0a isoforms did not show evident WPB localization as observed in V0a1 and V0a2 (*Figure 1—figure supplement 6A,B*), we concluded that V0a1 and V0a2 are the predominant V0a isoforms located on WPBs in HUVECs.

## V0a1 and V0a2 are dispensable for VWF multimerization

As bafilomycin A1 and concanamycin A bind to V0c, a common subunit of the V-ATPase complex (*Bowman et al., 2006*; *Huss et al., 2002*; *Wang et al., 2021*), they are expected to block 'pan' V-ATPase independent of the variations of subunits/isoforms. To test whether cellular V-ATPase activity is required for the maturation of VWF, we examined the effects of V-ATPase inhibitors on VWF multimerization. VWF is secreted at a low level, even in the absence of exogenous agonist stimulation. The secretion occurs from the WPBs and the TGN (referred to as 'basal' and 'constitutive' secretions, respectively). Although the secretion is at a low level, it is thought that the continuous supply of VWF multimers via these pathways is crucial for the maintenance of plasma VWF levels for the immediate hemostatic response (*Giblin et al., 2008*). While V-ATPase blockage did not change the amount of basally/constitutively secreted VWF (*Figure 1—figure supplement 2B*), the multimer analysis revealed that it caused a selective induction of dimer secretion (*Figure 2—figure supplement 1A*), suggesting a requirement for cellular V-ATPase activity in multimerization. As the dimer secretion is blocked by brefeldin A, an ER-to-Golgi transport inhibitor (*Figure 2—figure supplement 1B*), it is likely that the dimer secretion caused by V-ATPase blockage occurs from the TGN via the constitutive secretion pathway. The V-ATPase inhibitor exposure also caused a rapid disappearance of the nascent WPB buds (*Figure 2—figure supplement 1C*, compared with *Figure 2B*). The above observations suggested that VWF multimerization occurring at the TGN and/or in nascent WPB buds require V-ATPase activity. Next, we tested whether the V0a isoforms found in WPBs contribute to multimerization. Although the depletion of either V0a1 or V0a2 caused an increase in the secreted amount of VWF multimers, unexpectedly, no detectable alteration was observed in the multimer status even when both isoforms were depleted (*Figure 3—figure supplement 1A,B*). V-ATPase contributes to the formation of intracellular acidic microenvironment

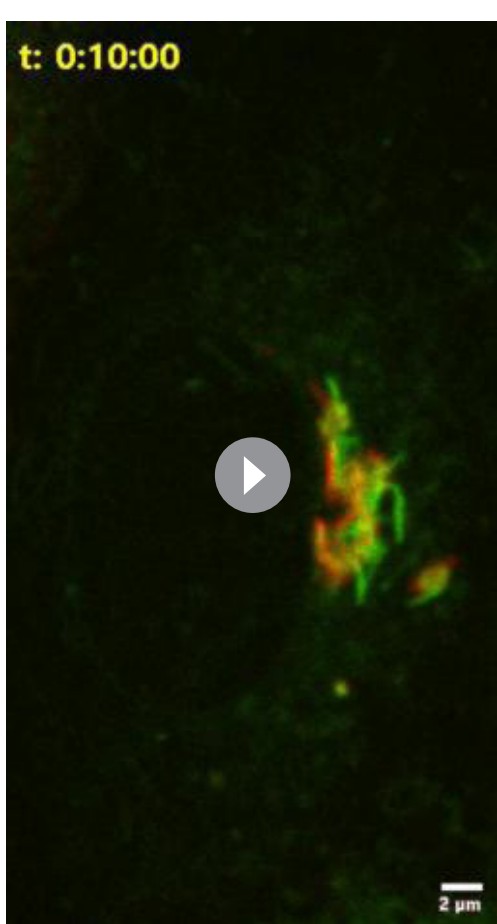

**Video 1.** V0a2-EGFP is found on bud of newly forming Weibel–Palade bodies. V0a2-EGFP (green) disappeared as soon as it segregates from the Golgi apparatus (red, mKate2-Giantin).
https://elifesciences.org/articles/71526/figures#video1

in various cell types (*Collins and Forgac, 2020*; *Maxson and Grinstein, 2014*; *Nakanishi-Matsui et al., 2010*; *Vasanthakumar and Rubinstein, 2020*). We introduced ratiometric pHluorin coupled with beta-1,4-galactosyltransferase 1 (GalT-pHluorin) (*Llopis et al., 1998*; *Cole et al., 1996*) to see if the V0a isoforms are involved in the luminal pH regulation of the TGN where WPBs are formed. GalT-pHluorin is able to measure pH in a range of 5.5–7.5 (*Figure 3—figure supplement 2A*). Bafilomycin A1 and concanamycin A exposure caused almost complete neutralization of the TGN, whereas the depletion of each V0a isoform resulted in a modest increase of the pH (*Figure 3—figure supplement 2A*). These observations suggest that the luminal acidic environment of the TGN is largely maintained by V-ATPase activity in HUVECs, but the contribution of V0a1 and V0a2 is limited or masked by functional redundancies. Consistent with the nascent WPB/TGN localization, V0a2 depletion resulted in a slight but homogenous increase of the luminal TGN pH (*Figure 3—figure supplement 2B*). In contrast, although no statistical difference was observed when the whole TGN pH was compared (*Figure 3—figure supplement 2A*), V0a1 depletion caused relatively large pH increase only in the some TGN subdomains (arrowheads in *Figure 3—figure supplement 2B*); suggesting an involvement of V0a1 in the pH regulation of the TGN, although the main cellular pool is found on peripheral WPBs. Taken together, despite the considerable localization on WPBs as well as the partial contribution on the acidic microenvironment of the TGN, V0a1 and V0a2 appear to be dispensable for multimer formation; however, the roles of these V0a isoforms remain elusive.

## The V0a isoforms are required for the biogenesis of WPBs

An increase in basal/constitutive VWF secretion is often observed in cells showing an alteration in biogenesis of WPBs (*Nightingale et al., 2009*; *Lui-Roberts et al., 2005*). To evaluate the importance of the V0a isoforms in the biogenesis of WPBs, we examined WPBs in V0a-depleted cells (*Figure 3—figure supplement 1A*). The WPBs were visualized using an anti-VWF antibody. Depletion of V0a2, which is found in the nascent WPBs, resulted in the formation of irregular, twisted WPBs compared to those found in the control (*Figure 3A*). To clarify the morphological change of WPBs, we determined the longest Feret's diameter ($F_{max}$) of individual WPBs and the aspect ratio ($F_{max}/F_{min}$) calculated from the longest and shortest Feret's diameters. Both values are expected to be smaller if WPBs are twisted or in irregular shape compared with relatively straight structure observed in the control. Accordingly, Feret's diameters of WPBs, particularly $F_{max}/F_{min}$ in the V0a2-depleted cells were obviously smaller than those in the control (*Figure 3—figure supplement 3*). To confirm the shRNA-mediated phenotype, we introduced HA-tagged shRNA-resistant V0a2 in shRNA-expressing cells. The phenotype was specifically rescued in the resistant V0a2 expressing cells; however, this was not observed in the non-transfected cells (*Figure 3B*). The number of WPBs formed in the V0a2-depleted cells was comparable to that in the control (*Figure 4B*), suggesting that V0a2 is not essential for the separation of WPBs from the TGN at the site of biogenesis. WPBs are storage organelles; that is, once formed, they remain in cells until they are exocytosed or turned over. Efficient reduction of target proteins by shRNAs requires time. Thus, it is likely that immunofluorescence of shRNA-transfected cells using anti-VWF antibody detects not only the WPBs that are formed in the absence of the target protein, but also the pre-formed WPBs before depletion. To further confirm the significance of V0a2 in biogenesis of WPBs, HUVECs were transduced with recombinant lentiviral particles expressing HA-tagged VWF after confirmed the depletion of endogenous V0a2 protein (*Figure 3—figure supplement 1C*). VWF$^{D1-A1}$-HA was found in the irregularly shaped WPBs located away from the TGN (*Figure 3C*). The WPBs were decorated with EGFP-Rab27a (*Figure 3D*). These observations indicate that the membrane-bound structure of WPBs can still be formed and transported to the cell periphery, even in the absence of V0a2, although it appears to be important for the formation of properly shaped WPBs.

The phenotype induced by the depletion of V0a1, which is found in the peripheral mature WPBs, is more striking. The number of WPBs in the V0a1-depleted cells was substantially decreased compared to that in the control, and the formed WPBs were largely accumulated around the TGN (*Figure 4A–B*). Large decrease in Feret's diameter in the V0a1-depleted cells (*Figure 3—figure supplement 3*) seemed to represent the appearance of rounded WPBs observed in cell periphery (*Figure 4A*), although the WPBs accumulated around the TGN still retain the characteristic elongated shape. As the rounded WPBs were not often observed when WPBs were visualized by the introduction of VWF$^{D1-A1}$-EGFP and P-selectin-EGFP in the V0a1-depleted cells (*Figure 4D and E*), we conclude that they are WPBs formed before the V0a1 deletion was achieved. The perinuclear accumulation of WPBs was

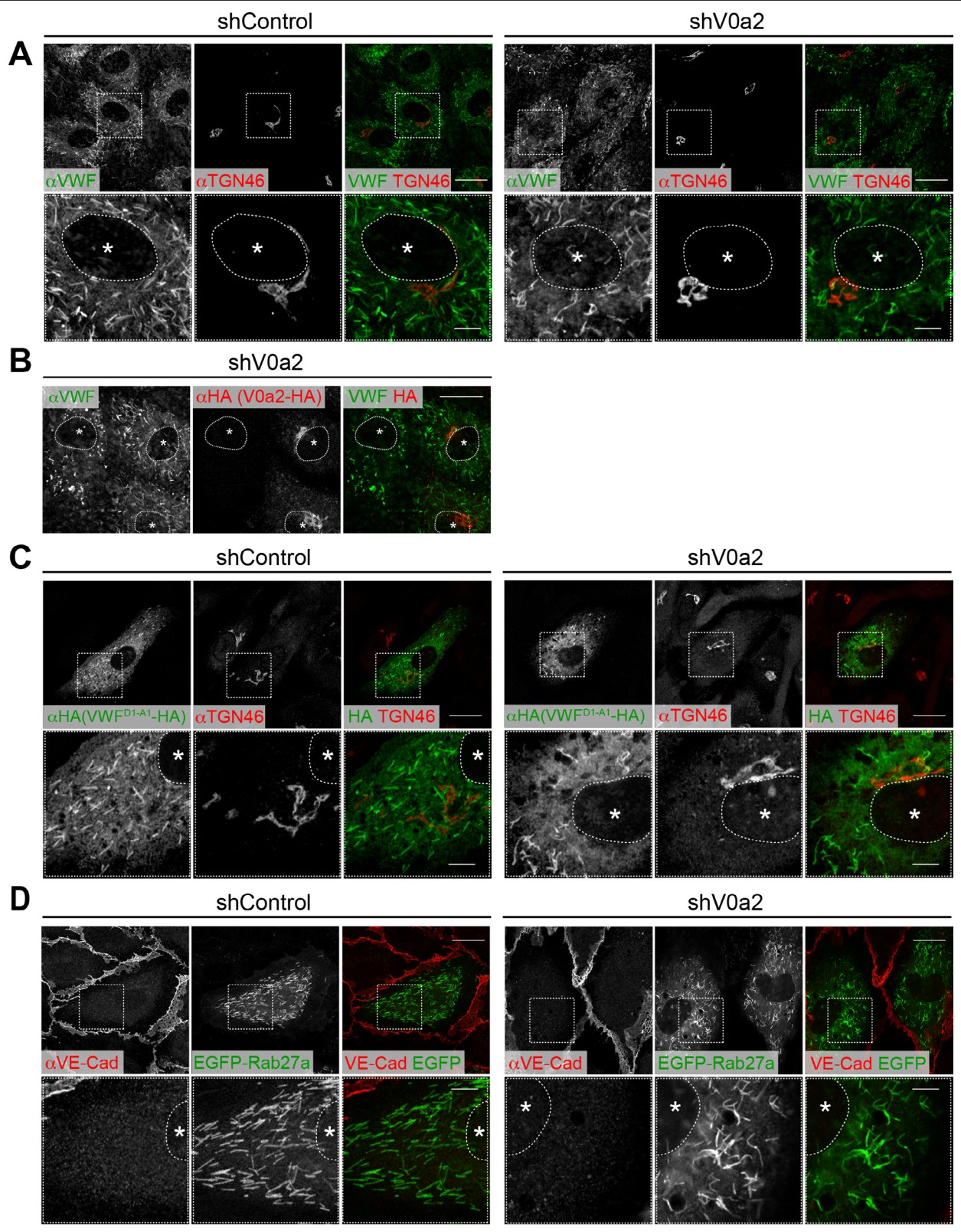

**Figure 3.** Depletion of V0a2 causes the formation of irregular, twisted Weibel–Palade bodies (WPBs). (**A**) Irregular WPBs were generated in the V0a2-depleted cells. WPBs were visualized using immunofluorescence with anti-von Willebrand factor (VWF) antibody. Scale bars, 20 μm (wide field) and 5 μm (expanded). (**B**) The irregular WPBs were rescued by the induction of shRNA-resistant V0a2 (V0a2-HA). Scale bar, 20 μm. (**C, D**) WPBs separated from the *trans*-Golgi network and maturated even in the absence of V0a2. VWF$^{D1-A1}$-HA (**C**) and EGFP-Rab27a (**D**) were introduced in the V0a2-depleted

*Figure 3 continued on next page*

Figure 3 continued

human umbilical vein endothelial cells. Due to limitations in the size for packaging into a lentivirus particle, we used VWF$^{D1-A1}$ instead of full-length VWF. VWF$^{D1-A1}$ is shown to function in the same manner as full-length VWF in cells (*Michaux et al., 2006*). Scale bars, 20 µm (wide field) and 5 µm (expanded). Asterisks, nucleus.

The online version of this article includes the following source data and figure supplement(s) for figure 3:

**Figure supplement 1.** Depletion of the V0a isoforms does not alter von Willebrand factor (VWF) multimerization.

**Figure supplement 1—source data 1.** Uncropped immunoblot images for *Figure 3—figure supplement 1A, B and C*.

**Figure supplement 2.** V-ATPase blockage causes almost total neutralization of the luminal TGN pH, while the depletion of the V0a isoforms results in a modest pH increase.

**Figure supplement 2—source data 1.** Graph data for *Figure 3—figure supplement 2A*.

**Figure supplement 3.** Frequency distribution of the longest Feret's diameter ($F_{max}$) and the aspect ratio of the longest and shortest distance ($F_{max}/F_{min}$) of individual WPBs in the V0a-depleted cells.

**Figure supplement 3—source data 1.** Graph data for *Figure 3—figure supplement 3*.

rescued by the induction of shRNA-resistant V0a1 (*Figure 4C*). The phenotype was further confirmed by the induction of tagged VWF$^{D1-A1}$ and P-selectin in V0a1-depleted cells (*Figure 4D–E*). P-selectin is a leukocyte-recruiting protein that is packaged into WPBs at the TGN (*Harrison-Lavoie et al., 2006*; *Larsen et al., 1989*). Both proteins were trapped in the perinuclear buds originating from the TGN in the absence of V0a1 (*Figure 4D–E*). The accumulated WPB buds were not EGFP-Rab27a-positive (*Figure 4F*). Live cell imaging showed that the accumulated WPBs in the V0a1-depleted cells were rather static compared to those in the control (*Videos 2 and 3*). These observations suggest that newly forming WPBs cannot be separated from the TGN in the absence of V0a1. Post-budded WPBs from the TGN are transported along the microtubule network to the cell periphery, where they are tethered to actin fibers (*Bierings et al., 2012*; *Nightingale et al., 2009*; *Rojo Pulido et al., 2011*). We believe that disruption of cytoskeletal architecture is unlikely because we were unable to observe any apparent alteration of microtubule and actin networks in the V0a1-depleted cells (*Figure 4— figure supplement 1A*). Accordingly, the induction of a dominant-negative form of either dynactin 2 (*Rondaij et al., 2006*) or myosin Va (*Rojo Pulido et al., 2011*), which disturbs the microtubule or actin network-mediating transport of WPBs, did not mimic the phenotype upon V0a1-depletion (*Figure 4—figure supplement 1B,C*).

## Protein kinase D, a membrane fission regulator is required for the biogenesis of WPBs

At the TGN, newly produced proteins that transition through the Golgi apparatus are sorted and packaged into distinct membrane-bound structures (*De Matteis and Luini, 2008*; *Kienzle and von Blume, 2014*; *Pakdel and von Blume, 2018*; *Anitei and Hoflack, 2011*). The formation is initiated by the deformation of the TGN membrane to form a bud where the proteins are packaged. The formed bud then segregates from the TGN membrane by membrane fission for further transport (*Campelo and Malhotra, 2012*; *Renard et al., 2018*). In the absence of V0a1, WPBs seemed unable to separate from the TGN, although the buds were formed (*Figure 4*). We postulated that V0a1 is required for membrane fission of WPBs at the TGN. Thus, we screened known TGN membrane fission regulators, such as dynamins (*Ferguson and De Camilli, 2012*; *Hinshaw, 2000*), CtBP1/BARS (*Bonazzi et al., 2005*; *Pagliuso et al., 2016*; *Valente et al., 2012*; *Weigert et al., 1999*), and serine/threonine-protein kinases PKDs (*Bard and Malhotra, 2006*; *Bossard et al., 2007*; *Liljedahl et al., 2001*; *Malhotra and Campelo, 2011*; *Van Lint et al., 2002*), to determine whether any of these contribute to the biogenesis of WPBs. Although the dominant-negative forms of various splicing variants of dynamin 2, dynamin 3, and CtBP1/BARS are virtually ineffective in biogenesis of WPBs (*Figure 5—figure supplement 1*, *Figure 5—figure supplement 2*), we found that the induction of the kinase-dead mutants of the PKD family members (PKD1$^{K612N}$, PKD2$^{K580N}$, and PKD3$^{K605N}$) in HUVECs significantly impaired the formation of WPBs (*Figure 5A*, *Figure 5—figure supplement 3A,B,C*). The expression of kinase-dead PKD mutants leads to the formation of tubules originating from the TGN because of the blockage of PKD-mediated membrane fission. In such cells, cargo proteins that should be transported in a PKD-dependent manner are trapped in the tubules (*Liljedahl et al., 2001*). Endogenous VWF protein was found in the tubules formed by all three PKD mutants (*Figure 5A*, *Figure 5—figure supplement*

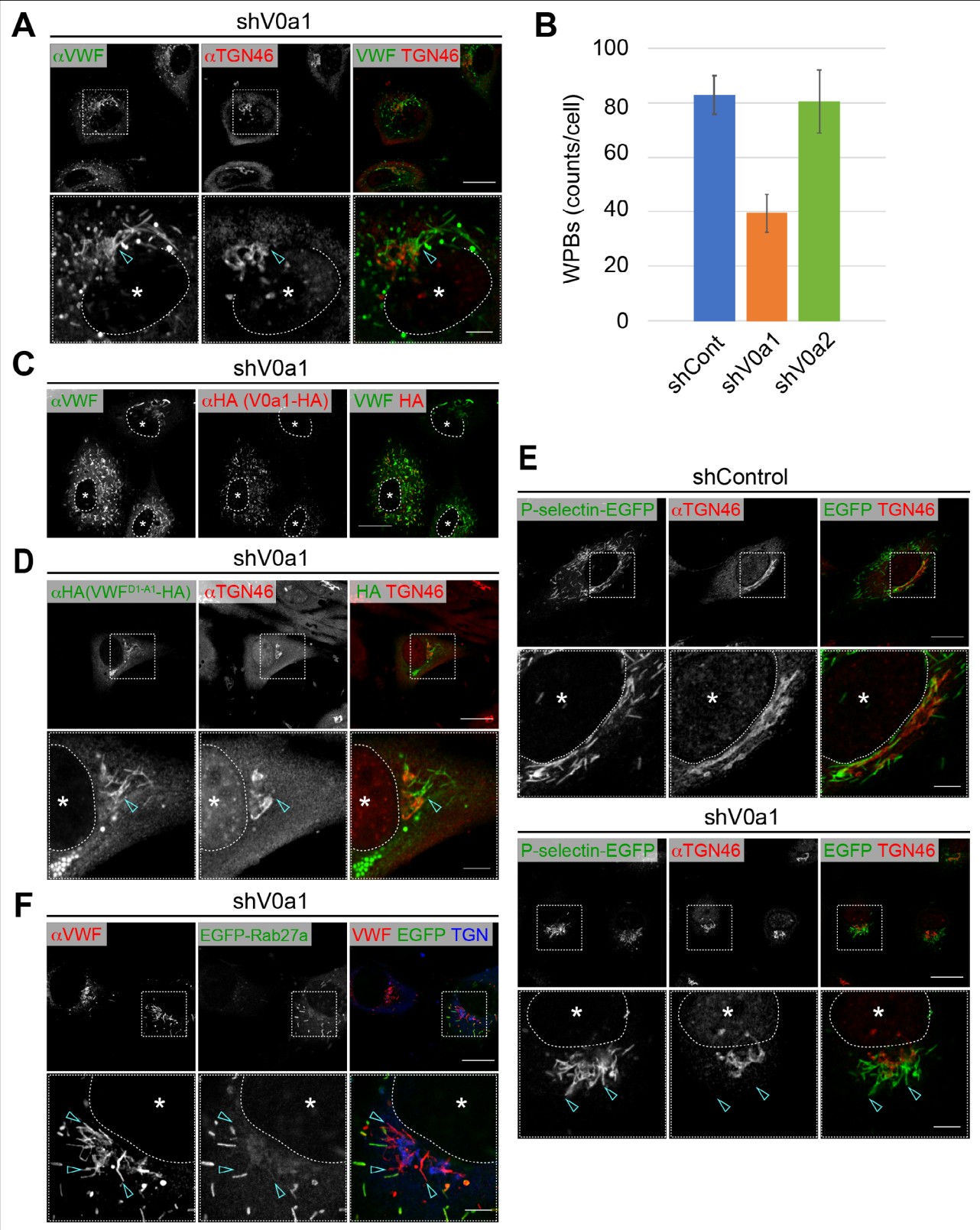

**Figure 4.** Weibel–Palade bodies (WPBs) cannot be separated from the *trans*-Golgi network in the absence of V0a1. (**A**) V0a1 depletion caused the perinuclear accumulation of WPBs (arrowheads, compared to that in the shControl in *Figure 3A*). Scale bars, 20 µm (wide field) and 5 µm (expanded). (**B**) The depletion of V0a1 resulted in the significant decrease in the number of WPBs (counts per cell, average± SD). We confirmed that there are no obvious difference in cell size between the V0a isoform depleted cells and the control. n = 30 (shControl), n = 29 (shV0a1), and n = 31 (shV0a2).

*Figure 4 continued on next page*

*Figure 4 continued*

Eight independent fields were collected from each imaging sample. The images were analyzed using Fiji. *p < 0.05 to shControl. (C) The phenotype mediated by V0a1 depletion was rescued by the induction of a shRNA-resistant V0a1 (HA-V0a1). Scale bar, 20 µm. (D, E) WPB contents were trapped in the perinuclear WPBs that accumulated in the absence of V0a1. VWF$^{D1-A1}$-HA (D) and P-selectin-EGFP (E) were transfected in V0a1-depleted cells. Both proteins were observed in the accumulated WPBs (arrowheads, compared to that in the shControl in *Figure 3C* for VWF$^{D1-A1}$-HA). (F) EGFP-Rab27a was absent in perinuclear WPBs that accumulated in V0a1-depleted cells (arrowheads). Scale bars, 20 µm (wide field) and 5 µm (expanded). Asterisks, nucleus. Source files of all graph data shown in this figure are available in *Figure 4—source data 1*.

The online version of this article includes the following source data and figure supplement(s) for figure 4:

**Source data 1.** Graph data for *Figure 4B*.

**Figure supplement 1.** Microtubule and actin networks are unaffected by depletion of V0a isoforms.

*3A,B*), indicating that the membrane fission of WPBs at the TGN requires PKDs. We further investigated whether V0a1 had any effect on the subcellular localization of wild-type PKDs (PKDs$^{WT}$). EGFP-PKD1$^{WT}$ was found mainly in the cytosol and certain parts of the TGN in the control cells, as reported earlier (*Figure 5B*; *Bard and Malhotra, 2006*; *Malhotra and Campelo, 2011*; *Van Lint et al., 2002*). EGFP-PKD2$^{WT}$ and EGFP-PKD3$^{WT}$ showed comparable localization to EGFP-PKD1$^{WT}$ (*Figure 5—figure supplement 3D,E*). In contrast, EGFP-PKDs$^{WT}$ substantially accumulated on the WPBs, which were retained around the TGN upon V0a1 depletion (*Figure 5B–E*, *Figure 5—figure supplement 3D,E*). The induction of PKDs$^{WT}$ appears to be not enough to induce the separation of WPBs from the TGN in the V0a1-depleted cells. A similar accumulation of endogenous PKD2 protein was observed in the V0a1-depleted cells (*Figure 5—figure supplement 4*). Thus, the above observations indicate that the segregation of newly forming WPB buds from the TGN is mediated by PKD-dependent membrane fission, and suggest that either V0a1 is required for this process or that V0a1 and PKD coordinate with each other in this process.

## Diacylglycerol and phosphatidylinositol-4-phosphate are present on nascent WPBs

To further elucidate the underlying mechanisms of V0a1-mediated regulation of membrane fission, we investigated how PKDs could be recruited to newly forming WPBs. Subcellular localization of PKDs is regulated by diacylglycerol (DAG), a membrane lipid (*Baron and Malhotra, 2002*; *Maeda et al., 2001*). We used the C1 domain from protein kinase Cε (EGFP-PKCε-C1) as a high-affinity DAG reporter (*Lehel et al., 1995*; *Stahelin et al., 2005*). A large DAG pool was detected on the TGN, as expected from an earlier study (*Figure 6A*;

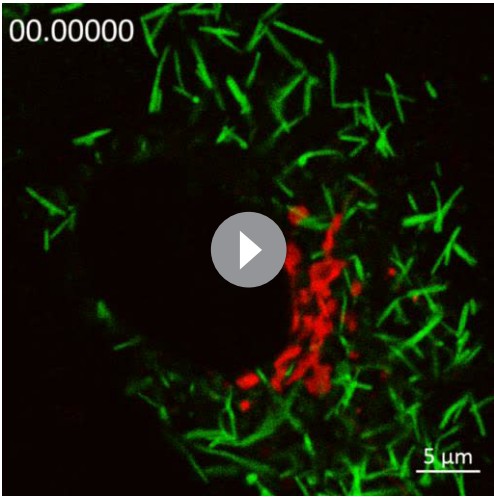

**Video 2.** Weibel–Palade bodies (WPBs) in the control shRNA introduced cells. WPBs were marked with P-selectin-EGFP (green), and the Golgi apparatus was identified using mKate2-Giantin (red). shControl, Video 2; shV0a1, Video 3.

https://elifesciences.org/articles/71526/figures#video2

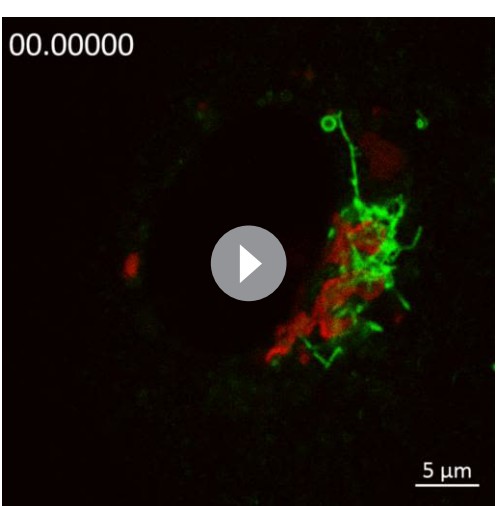

**Video 3.** Accumulated Weibel–Palade bodies (WPBs) at the Golgi apparatus in the V0a1-depleted cells are static compared to those in the control.

https://elifesciences.org/articles/71526/figures#video3

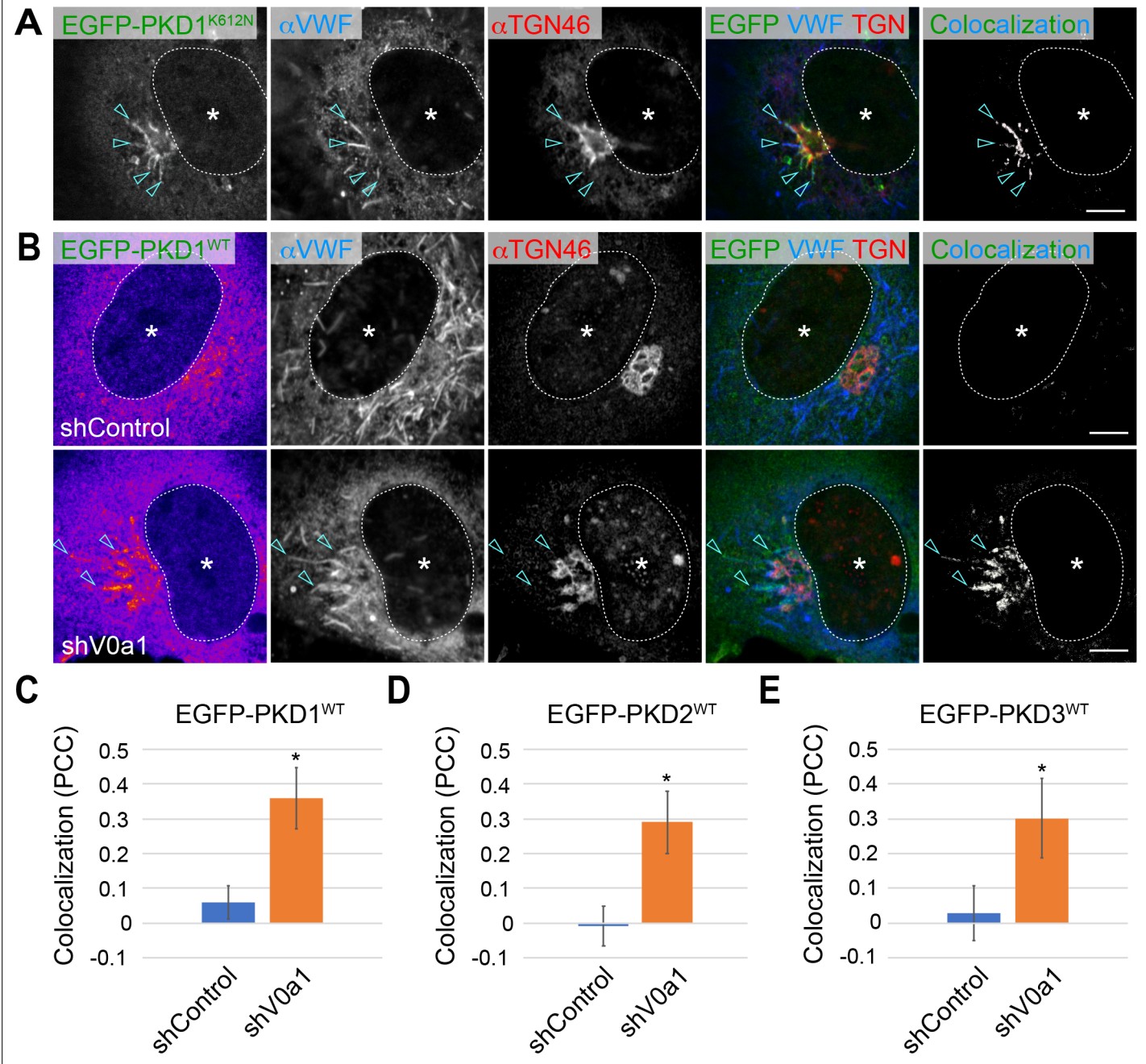

**Figure 5.** Weibel–Palade bodies (WPB) biogenesis requires protein kinase D1 (PKD1). (**A**) The von Willebrand factor (VWF) was trapped in the tubules (arrowheads) formed by the induction of a dominant-negative form of PKD1 (EGFP-PKD1$^{K612N}$) because of the blockage of PKD1-mediating membrane fission at the TGN (arrowheads). (**B**) EGFP-PKD1$^{WT}$ was accumulated on the perinuclear WPBs that accumulated in the absence of V0a1. The channel overlap of EGFP-PKD1$^{WT}$ and VWF are shown as 'colocalization' on the right. Scale bars, 5 μm. **C–E.** The localization of EGFP-PKDs on WPBs was significantly increased in the V0a1-depleted cells. The colocalization of VWF and EGFP-PKD1$^{WT}$ (**C**), EGFP-PKD2$^{WT}$ (*Figure 5—figure supplement 3D*), and EGFP-PKD3$^{WT}$ (*Figure 5—figure supplement 3E*) is shown as Pearson's correlation coefficient (PCC, average± SD). n = 14 (VWF and EGFP-PKD1$^{WT}$), n = 12 (VWF and EGFP-PKD2$^{WT}$), and n = 13 (VWF and EGFP-PKD3$^{WT}$). *p < 0.05 to shControl. Asterisks, nucleus. Source file of the graph data shown in C–E is available in *Figure 5—source data 1*.

The online version of this article includes the following source data and figure supplement(s) for figure 5:

**Source data 1.** Graph data for *Figure 5C–E*.

**Figure supplement 1.** Induction of dominant-negative forms of dynamins does not alter the biogenesis of Weibel–Palade bodies.

**Figure supplement 2.** Induction of dominant-negative forms of CtBP1/BARS does not alter the biogenesis of Weibel–Palade bodies.

*Figure 5 continued on next page*

*Figure 5 continued*

**Figure supplement 3.** Biogenesis of Weibel–Palade bodies (WPBs) requires protein kinase D (PKD).

**Figure supplement 3—source data 1.** Graph data for *Figure 5—figure supplement 3C*.

**Figure supplement 4.** Endogenous protein kinase D2 (PKD2) protein accumulates in the *trans*-Golgi network (TGN) in the absence of V0a1.

*Lehel et al., 1995*). The reporter was also found in the perinuclear WPB buds in the control group (*Figure 6A*). The signal was more evident in the accumulated WPB buds formed upon V0a1-depletion (*Figure 6A*). DAG localization on perinuclear WPBs was further confirmed using two additional DAG reporters, including the C1a domain from PKD1 (*Figure 6B*, *Figure 6—figure supplement 1A*; *Baron and Malhotra, 2002*; *Maeda et al., 2001*; *Tsai et al., 2014*). These observations showed that DAG was present on the membrane of newly forming WPBs, and PKDs were recruited through the interaction with DAG. We further investigated whether the PKDs on the WPB buds were enzymatically active. PI4-KIIIβ, a type III phosphatidylinositol-4-kinase localized at the TGN, has been identified as a PKD substrate (*Hausser et al., 2005*). Its phosphorylation on the activation loop causes the activation of the enzyme, resulting in the conversion of phosphatidylinositol to phosphatidylinositol-4-phosphate (PI4P). We introduced the PH domain from the oxysterol-binding protein (EGFP-PH-OSBP) as a PI4P reporter in HUVECs (*Balla et al., 2005*). As observed in the DAG reporters, the signal was detected on the accumulated WPBs in the V0a1-depleted cells and on the perinuclear WPBs in the control to a lesser extent (*Figure 6C*). Two additional PI4P reporters, PH-FAPP1-EGFP (*Balla et al., 2005*) and EGFP-P4M-SidM (*Hammond et al., 2014*), also showed similar results (*Figure 6—figure supplement 1B,C*), indicating a presence of PI4P on perinuclear newly forming WPBs. To see if the PI4P on the forming WPBs is provided through the activation of PKDs, we performed western blotting of HUVEC lysate by using an anti-phospho-PKD1 antibody that detects the phosphorylation on the activation loop (pSer$^{744/748}$). V0a1-depletion mediated by two independent shRNAs caused an increase of total PKD1 protein compared with the control, whereas we were unable to see any detectable changes in the level of phosphorylated PKD1 (*Figure 6—figure supplement 2*). As the substantial amount of PKDs were observed on the accumulated WPBs (*Figure 5B–E*, *Figure 5—figure supplement 3D,E*), an increase of PKD1 protein might be due to a difference in the protein stability dependent on the subcellular localization. Alternatively, it may be due to an upregulation of PKD1 expression by a cell autonomous compensation or unknown mechanism mediated by the V0a1-depletion. Because of the low endogenous expression of PKDs, it is hard to test whether the PKDs observed on the accumulated WPBs in the V0a1-depleted cells are active/phosphorylated or not. Therefore, although total amount of cellular active PKD1 is unchanged independent from the absence or presence of V0a1, we were unable to clarify the requirement of V0a1 on the PKDs activation/phosphorylation.

## Discussion

Membrane fusion and fission are fundamental subcellular events essential for life. In contrast to membrane fusion, which is mainly driven by SNARE proteins and cofactors (*Südhof and Rothman, 2009*), the process of membrane fission is less understood (*Campelo and Malhotra, 2012*; *Renard et al., 2018*). Our study reveals that the V-ATPase subunit V0a1 promotes the biogenesis of WPBs through the regulation of membrane fission (*Figure 7*). V-ATPase has been suggested to regulate membrane fusion by generating electrochemical gradients or as a structural component of the fusion machinery (*Merz, 2015*), whereas any involvement in membrane fission has not been described previously. Although we were unable to determine whether the requirement of V0a1 is associated with the proton pump function because of the pleiotropic effects of V-ATPase in biogenesis or multimerization of WPBs, we believe that the identification of V0a1 as a new membrane fission regulator should facilitate the elucidation of the molecular events that allow membrane fission. V0a1 is found on the peripheral mature WPBs in endothelial cells, whereas V0a2, the other V0a isoform, is only observed in nascent WPBs. Consistent with our findings, these V0a isoforms were detected in WPB proteomes (*Holthenrich et al., 2019*; *van Breevoort et al., 2012*). V0a2 appears to be required for the formation of proper, elongated shaped WPBs, whereas it is not essential for bud formation or membrane fission of WPBs despite the specific localization on newly forming WPBs. Nevertheless, the requirement of V0a1, which is mainly found in peripheral WPBs, for membrane fission that occurs at the perinuclear TGN is mystifying. V0a1 is also found on small intracellular vesicles in addition to WPBs;

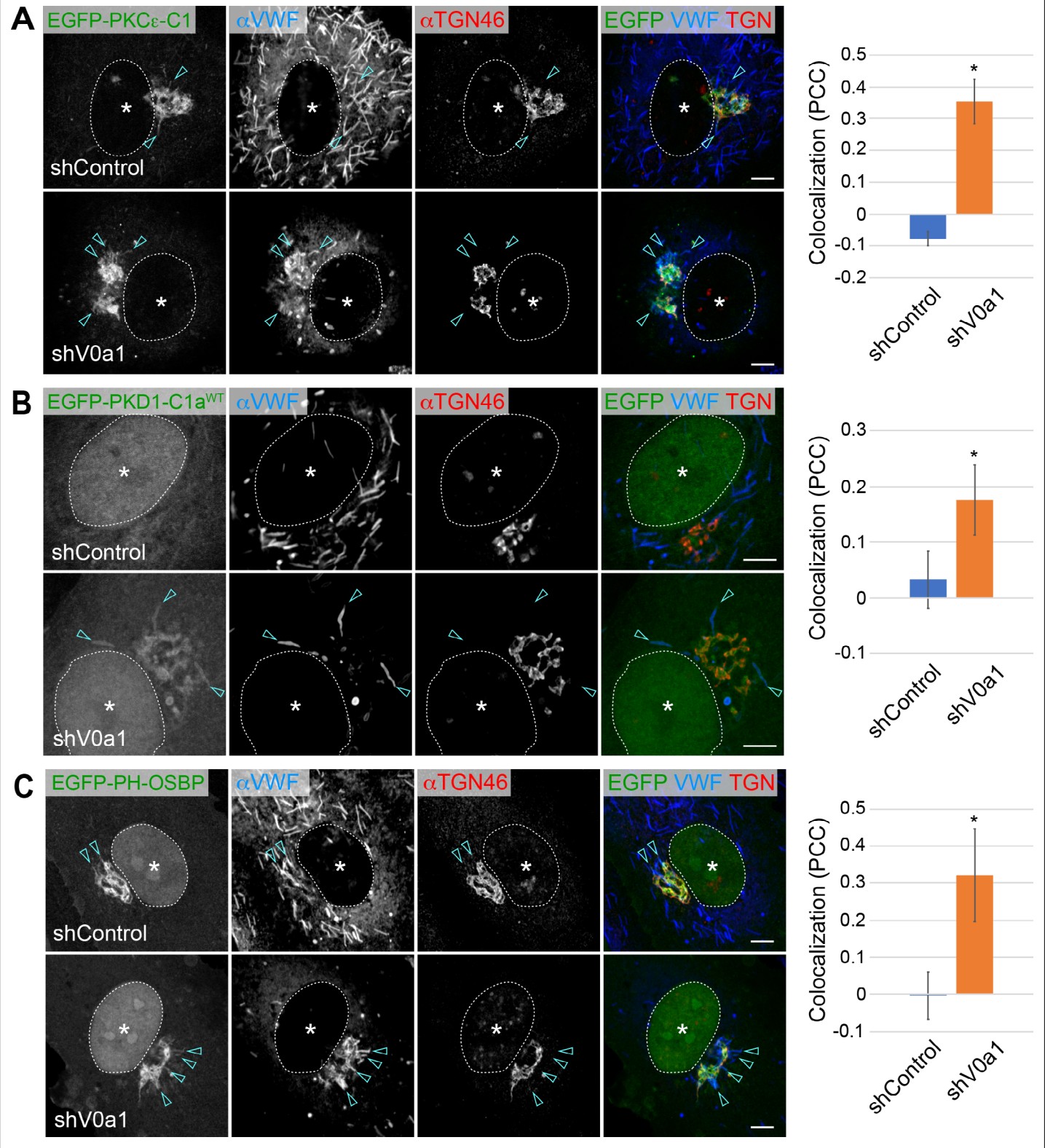

**Figure 6.** Diacylglycerol (DAG) and phosphatidylinositol-4-phosphate (PI4P) are present on newly forming Weibel–Palade bodies (WPBs). (**A, B**) DAG was present on newly forming perinuclear WPBs. Two specific DAG reporters, EGFP-PKCε-C1 (**A**) and EGFP-PKD1-C1a (**B**), were introduced into human umbilical vein endothelial cells. EGFP-PKCε-C1 (**A**) is found in the perinuclear WPBs of the control (arrowheads). The signal is more evident in the accumulated WPBs in V0a1-depleted cells (arrowheads). EGFP-PKD1-C1a (**B**) is found in the accumulated WPBs in V0a1-depleted cells (arrowheads). Since PKD1-C1a is not a high-affinity DAG reporter as PKCε-C1, it may explain why the signal is not evident in the control. The colocalization of von

*Figure 6 continued on next page*

*Figure 6 continued*

Willebrand factor (VWF) and EGFP-PKCε-C1 (**A**) and EGFP-PKD1-C1a (**B**) is shown as Pearson's correlation coefficient (PCC, average± SD) on the right. n = 19 (VWF and EGFP-PKCε-C1), and n = 14 (VWF and EGFP-PKD1-C1a). *p < 0.05 to shControl. (**C**) PI4P was present in perinuclear WPBs. EGFP-PH-OSBP, a PI4P reporter, was found in few perinuclear WPBs in the control (arrowheads). The signal is more evident in V0a1-depleted cells (arrowheads). The colocalization of VWF and EGFP-PH-OSBP is shown as Pearson's correlation coefficient (PCC, average ± SD) on the right. n = 18 (VWF and EGFP-PH-OSBP). *p < 0.05 to shControl. Scale bars, 5 µm. Asterisks, nucleus. Source file of the graph data shown in 6 A–C is available in *Figure 6—source data 1*.

The online version of this article includes the following source data and figure supplement(s) for figure 6:

**Source data 1.** Graph data for *Figure 6A–C*.

**Figure supplement 1.** Diacylglycerol (DAG) and phosphatidylinositol-4-phosphate (PI4P) are present on newly forming Weibel–Palade bodies (WPBs).

**Figure supplement 2.** Active/phosphorylated PKD1 is unchanged independent from the absence or presence of V0a1, whereas V0a1-depletion causes an increase of total PKD1 protein.

**Figure supplement 2—source data 1.** Uncropped immunoblot images for *Figure 6—figure supplement 2*.

perhaps, these are acidic subcellular compartments such as endosomes and lysosomes. Endosomes are observed to form close contacts with WPBs in electron tomographic analysis (*Berriman et al., 2009*). The endosomal/lysosomal tetraspanin CD63/LAMP3 is delivered to WPBs during maturation (*Harrison-Lavoie et al., 2006*). Thus, it is likely that the small vesicle-mediated transfer of V0a1 could drive the membrane fission of nascent WPBs. Relatively large pH increase was observed in some TGN subdomains in the V0a1-depleted cells (*Figure 3—figure supplement 2B*) possibly suggests that the biogenesis of WPBs occurs at particular subdomains of the TGN. Further work is required to elucidate the role of V0a1 in membrane fission.

We also found that DAG is present on the membrane of newly forming WPBs; however, it was not found in the post-budded WPBs. DAG accumulation on nascent WPBs became more evident in the absence of V0a1. Thus, PKDs appear to be recruited to the WPB through the interaction of the DAG-binding domain. Insertion of the N-terminal hydrophobic stretch of PKDs into the outer leaflet of the TGN is suggested to assist membrane fission by generating membrane curvature (*Campelo and Malhotra, 2012*; *Malhotra and Campelo, 2011*). We believe that the hydrophobic stretch-mediating mechanism is unlikely in the biogenesis of WPBs because we observed essentially equivalent effects on all PKD members, including PKD3, which lacks the hydrophobic stretch in the structure (*Campelo and Malhotra, 2012*). Instead, our observations suggest functional redundancy of PKDs in

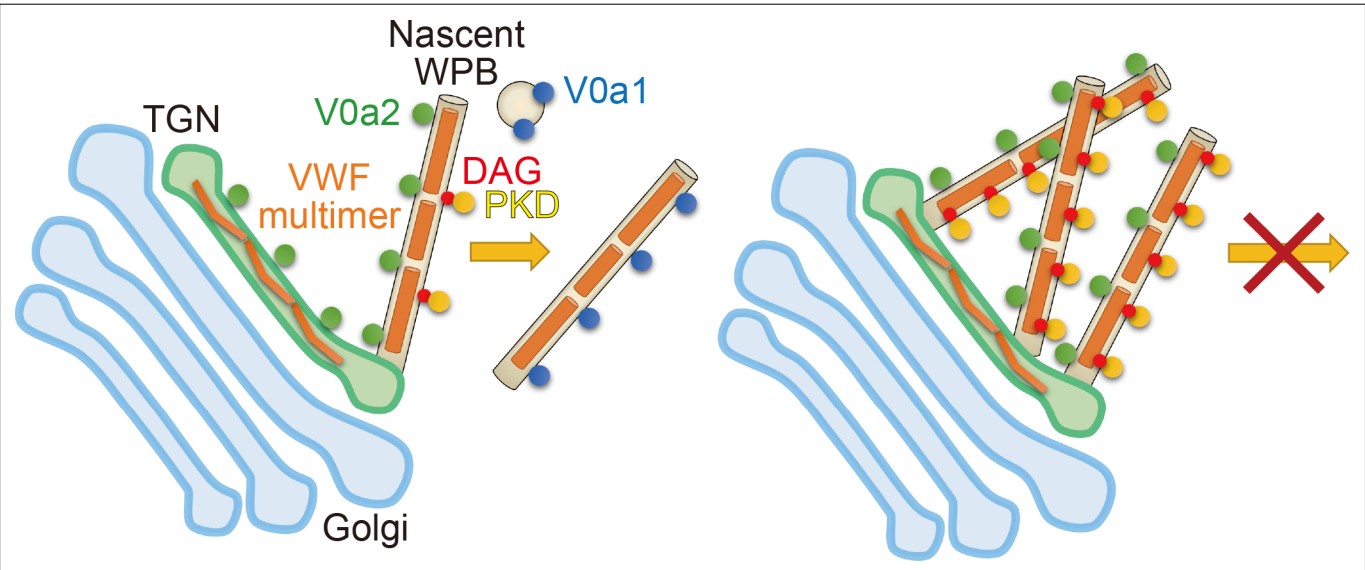

**Figure 7.** The biogenesis of Weibel–Palade bodies (WPBs) requires vacuolar H+-ATPase V0a1 and PKD for the membrane fission. V0a1 on small vesicles such as endosomes could drive the membrane fission (see *Discussion*). In the absence of V0a1, nascent WPBs could not separate from the *trans*-Golgi network (right). PKD activity was also required for the biogenesis of WPBs. Since the induction of wild-type PKDs did not support the segregation of WPBs from the TGN in the absence of V0a1, V0a1 appeared to be a primary factor in the membrane fission of WPBs.

biogenesis of WPBs. In fact, a single knockdown of either PKD1 or PKD2 does not show any significant effect on WPB formation as well as the secretion in earlier reports (*Ketteler et al., 2017*; *Hao et al., 2009*). DAG is a conical-shaped lipid that is thought to be relatively concentrated in the TGN buds to increase membrane curvature (*Bankaitis et al., 2012*). A theoretical model suggests that the presence of DAG on buds can stabilize the structure and prevent membrane fission (*Shemesh et al., 2003*). The disappearance of DAG in the post-budded WPBs may suggest a conversion of DAG into other lipids that destabilize the membrane structure to enable membrane fission. We also observed PI4P, a well-characterized membrane signaling lipid, on perinuclear nascent WPBs similar to DAG. PI4P and phosphatidic acid, a lipid generated from DAG through single-step metabolism, are shown to work as pH biosensors in cells; the pH-dependent interaction with particular proteins regulates cellular processes such as membrane biogenesis and intracellular trafficking (*Shin et al., 2020*; *Young et al., 2010*). TGN pH measurements suggest that V0a1 appears to regulate luminal pH in particular TGN subdomains in HUVECs (*Figure 3—figure supplement 2B*). Thus, the possible contribution of these lipids and cellular pH, as well as related lipid metabolism enzymes and lipid transporting proteins in WPB biogenesis should be investigated in the future.

In conclusion, we have shown that two V0a isoforms of V-ATPase promote the biogenesis of WPBs in endothelial cells. In particular, V0a1 is essential for the membrane fission in nascent WPBs at the TGN. We also found that PKD activity is required for WPB biogenesis. As the induction of wild-type PKDs in V0a1-depleted cells did not support WPB segregation from the TGN, V0a1 appears to be a primary factor in the membrane fission of WPBs, although the molecular link between V0a1 and PKDs is still unclear. Qualitative or quantitative deficiency of plasma VWF multimers causes a bleeding disorder called von Willebrand disease (*Sadler, 2005*; *Sadler, 1998*; *Sharma and Haberichter, 2019*). A V0a2 mutation has recently been reported to cause bleeding diathesis in humans (*Karacan et al., 2019*). Plasma VWF activity is significantly impaired in patients without any substantial loss of plasma VWF antigen level, thus exhibiting a typical sign of von Willebrand disease, although no associating mutation was found in the *VWF* gene. We believe that our findings will facilitate further understanding of the pathology associated with VWF, such as von Willebrand disease, as well as the mechanism of membrane fission.

## Materials and methods

Key resources table

| Reagent type (species) or resource | Designation | Source or reference | Identifiers | Additional information |
|---|---|---|---|---|
| Gene (*Homo sapiens*) | ATP6V0A1 | NA | Q93050 (VPP1_HUMAN) | V-type proton ATPase 116 kDa subunit a1 |
| Gene (*Homo sapiens*) | ATP6V0A2 | NA | Q9Y487 (VPP2_HUMAN) | V-type proton ATPase 116 kDa subunit a2 |
| Cell line (*Homo sapiens*) | Lenti-X 293T Cell Line | Clontech | Cat#632180 | |
| Cell line (*Homo sapiens*) | HUVEC | Lonza | Cat#C2519A | |
| Antibody | anti-VE-Cadherin (mouse monoclonal) | R&D Systems | Cat#123413 RRID:AB_2260374 | (1:1,000) |
| Antibody | anti-ZO-1 (rabbit monoclonal) | CST | Clone:D6L1E RRID:AB_2798287 | (1:50) |
| Antibody | anti-VWF (rabbit polyclonal) | Dako/Agilent Technologies | Cat#P0226 RRID:AB_579516 | (1:1,000) |
| Antibody | anti-VWF (sheep polyclonal) | Abcam | Cat#ab11713 RRID:AB_298501 | (1:500) |
| Antibody | anti-V0c (rabbit polyclonal) | Thermo-Fisher Scientific | Cat#PA5-66746 RRID:AB_2662731 | (1:25) |
| Antibody | anti-V1A (rabbit polyclonal) | Thermo-Fisher Scientific | Cat#PA5-65137 RRID:AB_2662833 | (1:125) |
| Antibody | anti-V0a1 (rabbit polyclonal) | Novus Biologicals | Cat#NBP1-89342 RRID:AB_11015740 | (1:200) |

*Continued on next page*

*Continued*

| Reagent type (species) or resource | Designation | Source or reference | Identifiers | Additional information |
|---|---|---|---|---|
| Antibody | anti-HA (mouse monoclonal) | Covance | Clone:16B12 RRID:AB_10064068 | (1:1,000) |
| Antibody | anti-TGN46 (sheep polyclonal) | Bio-Rad | Cat#AHP500GT RRID:AB_2203291 | (1:500) |
| Antibody | anti-V0a2 (rabbit polyclonal) | Atlas Antibodies | Cat#HPA044279 RRID:AB_10961274 | (1:50) |
| Antibody | anti-V0a2 (rabbit polyclonal) | My BioSource | Cat#MBS8527160 | (1:100) |
| Antibody | anti-α-tubulin (mouse monoclonal) | CST | Clone:DM1A RRID:AB_1904178 | (1:2,000) |
| Antibody | anti-PKD2 (rabbit monoclonal) | Abcam | Clone:EP1495Y RRID:AB_922233 | (1:200) |
| Antibody | anti-PKD2 (rabbit polyclonal) | GeneTex | Cat#C1C3 RRID:AB_1951437 | (1:200) |
| Antibody | anti-phospho-PKD/PKCμ(Ser744/448) (rabbit polyclonal) | CST | Cat#2054 RRID:AB_2172539 | (1:1,000) |
| Antibody | anti-PKD/PKCμ (rabbit monoclonal) | CST | Clone:D4J1N RRID:AB_2800149 | (1:1,000) |
| Antibody | Alexa 488-, 555-, 647 secondaries | Thermo Fisher Scientific | | (1:400) |
| Antibody | Cy3-secondary | Jackson Immunoresearch | | (1:400) |
| Recombinant DNA reagent | pLVSIN-CMV Pur Vector | Clontech | Cat#6183 | |
| Recombinant DNA reagent | Mission shRNA plasmid non-targeting control | Sigma-Aldrich | SHC016 | |
| Recombinant DNA reagent | Mission shRNA plasmid against V0a1 | Sigma-Aldrich | TRCN0000380234 | Mainly used throughout the study. |
| Recombinant DNA reagent | Mission shRNA plasmid against V0a1 | Sigma-Aldrich | TRCN0000333635 | Used in *Figure 6—figure supplement 2B*. |
| Recombinant DNA reagent | Mission shRNA plasmid against V0a2 | Sigma-Aldrich | TRCN0000043494 | |
| Chemical compound, drug | Bafilomycin A1 | Cayman Chemicals | Cat#11038 | 100 nM |
| Chemical compound, drug | iFluor555-conjugated phalloidin | Cayman Chemicals | Cat#20552 | (1:400) |
| Chemical compound, drug | Concanamycin A | AdipoGen Life Sciences | Cat#BVT-0237 | 100 nM |
| Chemical compound, drug | Brefeldin A | Thermo Fisher Scientific | Cat#B7450 | 100 ng/ml |
| Chemical compound, drug | Histamine | Sigma-Aldrich | Cat#H7250 | 100 μM |
| Chemical compound, drug | TransIT Lenti transfection reagent | Clontech | Cat#MIR6600 | |
| Chemical compound, drug | Lentiviral High Titer Packaging Mix | Clontech | Cat#6194 | |

### Reagents

Bafilomycin A1 and iFluor555-conjugated phalloidin were purchased from Cayman Chemicals (MI, USA). Concanamycin A was purchased from AdipoGen Life Sciences (CA, USA). Brefeldin A was purchased from Thermo Fisher Scientific (MA, USA). Histamine was procured from Sigma-Aldrich (MO, USA). The following primary antibodies were used: mouse monoclonal anti-VE-Cadherin (#123413, 1:1,000, R&D Systems, MN, USA), rabbit monoclonal anti-ZO-1 (D6L1E, 1:50, CST, MA, USA), rabbit polyclonal anti-VWF (P0226, 1:1,000, Dako/Agilent Technologies, CA, USA), sheep polyclonal anti-VWF (ab11713, 1:500, Abcam, MA, USA), rabbit polyclonal anti-V0c (PA5-66746, 1:25, Thermo-Fisher Scientific), rabbit polyclonal anti-V1A (PA5-65137, 1:125, Thermo-Fisher Scientific), rabbit polyclonal anti-V0a1 (NBP1-89342, 1:200, Novus Biologicals, CO, USA), mouse anti-HA (16B12, 1:1000, Covance, MI, USA), sheep anti-TGN46 (AHP500GT, 1:500, Bio-Rad, CA, USA), rabbit polyclonal anti-V0a2 (HPA044279, 1:50,

Atlas Antibodies, Bromma, Sweden), rabbit polyclonal anti-V0a2 (MBS8527160, 1:100, My BioSource, CA, USA), mouse monoclonal anti-α-tubulin (DM1A, 1:2000, CST), rabbit monoclonal anti-PKD2 (EP1495Y, 1:200, Abcam), rabbit polyclonal anti-PKD2 (C1C3, 1:200, GeneTex, CA, USA), rabbit polyclonal anti-phospho-PKD/PKCµ(Ser744/448) (#2054, 1:1000, CST), and rabbit monoclonal anti-PKD/PKCµ (D4J1N, 1:1,000, CST). The secondary antibodies used were Alexa 488, Alexa 555, and Alexa 647 (1:400, Thermo Fisher Scientific), and Cy3 (1:400, Jackson Immunoresearch, PA, USA).

## cDNA and plasmids

The cDNA used in this study were amplified from HUVEC cDNA, unless otherwise stated. AP1M1 and AP1G1 were provided by RIKEN BRC through the National Bio-Resource Project of MEXT, Japan. P4M-SidM was obtained from Dr. Tamas Balla (National Institute of Health, USA). For the expression of tagged proteins, we introduced EGFP, mKate2, or HA cDNA into the multiple cloning site of pLVSIN lentiviral vector (Clontech, CA, USA), such that the inserted protein was tagged at either the N- or C-terminus with a flexible linker (GGGGS ×2). The pLVSIN constructs used in this study were as follows: pLVSIN-EGFP, pLVSIN-N-EGFP-V0a1, pLVSIN-V0a1-C-EGFP, pLVSIN-N-HA-V0a1, pLVSIN-N-EGFP-V0a2, pLVSIN-V0a2-C-EGFP, pLVSIN-V0a2$^{shRNAresist}$-C-3× HA pLVSIN-N-EGFP-V0a3, pLVSIN-V0a3-C-EGFP, pLVSIN-N-EGFP-V0a4, pLVSIN-V0a4-C-EGFP, pLVSIN-N-EGFP-Rab27a, pLVSIN-AP1M1-C-EGFP, pLVSIN-AP1G1-C-EGFP, pLVSIN-VWF$^{D1-A1}$(Met$^1$-Pro$^{1480}$)-C-3× HA, pLVSIN-P-selectin-C-EGFP, pLVSIN-GalT(Met$^1$-Gln$^{61}$)-pHluorin, pLVSIN-Dynamin 3–1 $^{K44A}$-C-EGFP, pLVSIN-Dynamin 3–2 $^{K44A}$-C-EGFP, pLVSIN-Dynamin 2–2 $^{K44A}$-C-EGFP, pLVSIN-Dynamin 2–3 $^{K44A}$-C-EGFP, pLVSIN-Dynamin 2–4 $^{K44A}$-C-EGFP, pLVSIN-Dynamin 2–5 $^{K44A}$-C-EGFP, pLVSIN-CtBP1/BARS$^{WT}$-C-EGFP, pLVSIN-CtBP1/BARS$^{S147A}$-C-EGFP, pLVSIN-N-EGFP-PKD1$^{WT}$, pLVSIN-N-EGFP-PKD1$^{K612N}$, pLVSIN-N-EGFP-PKD2$^{WT}$, pLVSIN-N-EGFP-PKD2$^{K580N}$, pLVSIN-N-EGFP-PKD3$^{WT}$, pLVSIN-N-EGFP-PKD3$^{K605N}$, pLVSIN-N-EGFP-PKCε-C1, pLVSIN-N-EGFP-PKD1-C1a, pLVSIN-PKCγ-C1A-C1A-C-EGFP, pLVSIN-N-EGFP-PH-OSBP, pLVSIN-PH-FAPP1-C-EGFP, pLVSIN-N-EGFP-P4M-SidM(Thr$^{546}$-Lys$^{647}$), pLVSIN-dynactin 2-C-mKate2, pLVSIN-N-mKate2-myosin Va tail, and pLVSIN-N-mKate2-Giantin(Glu$^{3131}$-Leu$^{3259}$). Mission shRNA plasmids against V0a1 (TRCN0000380234, targeting the 3′-UTR region, and TRCN0000333635, targeting the coding region around Tyr$^{620}$-Met$^{627}$) and V0a2 (TRCN0000043494, targeting the coding region around Phe37 -Val55) and non-targeting control (SHC016) were purchased from Sigma-Aldrich. For the construction of HA-tagged shRNA-resistant V0a2, four nucleotides located in the middle of the targeting region were substituted with synonymous codons.

## Cell culture

Primary HUVECs (C2519A, Lonza, Basel, Switzerland) were cultured at 37 °C in EGM-2 supplemented with 2% fetal bovine serum (Lonza). All plates and coverslips used for HUVECs were freshly coated with fibronectin from human plasma (20 µg/mL; Fujifilm Wako, Osaka, Japan). Lenti-X293T cells (Clontech) were maintained in Dulbecco's modified Eagle's medium (DMEM, Fujifilm Wako) supplemented with 10% EqualFETAL (Atlas Biologicals, CO, USA) and 1 µg/mL penicillin and streptomycin (Fujifilm Wako) at 37 °C.

## Lentivirus particle preparation

Lenti-X293T cells (Clontech) were plated at a density of $6 \times 10^6$ cells in 90 cm$^2$ plates coated with soluble type I-C collagen (Kurabo, Osaka, Japan). After overnight culture, the pLVSIN lentiviral vector was transfected together with lentivirus packaging mix using TransIT Lenti transfection reagents according to the manufacturer's protocol (Clontech). Conditioned media were collected three times for 96 hr as recombinant lentivirus particles. The virus particles were then precipitated with PEG6000, resuspended in an appropriate amount of phosphate-buffered saline (PBS)/DMEM, and stored at −80 °C until use.

## Immunofluorescence and image analysis

HUVECs were seeded at 50,000 cells/well on fibronectin-coated coverslips in 24-well plates. After 16–18 hr of culture, the cells were transduced with recombinant lentivirus particles overnight. The medium was replaced daily. For the expression of tagged proteins on the shRNA background, cells were exposed to the virus particles that expressed the tagged protein after 48 hr of shRNA transfection. On day 6, the cells were fixed with 4% paraformaldehyde for 25 min at 37 °C. After rinsing

with PBS, cells were quenched with 50 mM NH₄Cl/PBS for 5 min, permeabilized, and blocked in PBS containing 0.3% Triton X-100% and 5% normal serum from the respective species for 30 min. The cells were incubated with primary antibodies in blocking buffer overnight at 4 °C. Primary antibodies were detected using the appropriate fluorescent secondary antibodies. Coverslips were mounted using Fluoromount-G (SouthernBiotech, AL, USA) and imaged using LSM880 (Plan Apochromat ×63 objective, NA 1.4, ZEISS, Oberkochen, Germany), FV3000 (UPSAPO60XO × 60 60 objective, NA 1.35, Olympus, Tokyo, Japan), and FV1000 (UPlanSApo ×60 objective, NA 1.35, Olympus) microscopes. For the super-resolution analysis (*Figure 2B* and *Figure 2—figure supplement 1C*), images were acquired with LSM880 with AiryScan (estimated lateral resolution: 180 nm). EGFP was excited with 488 nm line from an argon laser and collected with MBS 488/561/633 (acquisition band: 500–540 nm). Cy3 was excited with 561 nm line from a DPSS laser and collected with MBS 488/561/633 (acquisition band: 590–630 nm). AlexaFluor647 was excited with 633 nm line from a HeNe laser and collected with MBS 488/561/633 (acquisition band: 645–755 nm). The acquired images were processed with Zen Black software by performing filtering, deconvolution and pixel reassignment to enhance spatial resolution. Images were analyzed using Fiji to count the number and size (Feret's diameter) of WPBs and co-localization analysis Colocalization Finder for the generation of channel overlap (*Figure 5*, *Figure 5—figure supplements 3 and 4*); https://imagejdocu.tudor.lu/plugin/analysis/colocalizationfinder/start, and EzColocalization for the calculation of Pearson's correlation coefficient (*Figures 5 and 6*; https://github.com/DrHanLim/EzColocalization/releases/tag/1.1.4 *Sheng, 2020*) was performed. For the ratiometric pH measurement, GalT-pHluorin was consecutively excited at 405 and 488 nm and the emitted fluorescence was acquired through a spectral slit of 510–550 nm. Mean emission intensities obtained from the identical ROI were used to calculate 405/488 ratio. pH calibration standard was obtained by using monensin-containing calibration buffers adjusted to the indicated pH (*Grillo-Hill et al., 2014*; *Ma et al., 2017*). The ratiometric images were generated by the image calculator function on Fiji and the pH calibration data was applied to convert the ratio to pH.

## VWF multimer analysis

HUVECs were seeded and transduced using the same protocol used for immunofluorescence assay, except for the use of coverslips. On the day of analysis, cells were rinsed twice with endothelial basal medium (EBM)–2 without growth supplements or fetal bovine serum. After 2 hr of resting in EBM-2 containing 10 mM HEPES pH 7.4, conditioned media was collected for 1 hr to obtain basally/constitutively secreted VWF. The collected media was mixed with sodium dodecyl sulfate (SDS)–polyacrylamide gel electrophoresis sample buffer without reducing reagent and then loaded onto a 'SDS–agarose' gel. After separation, proteins on the gel were capillary blotted onto PVDF membranes and labeled with rabbit anti-VWF antibody (Dako/Agilent Technologies), followed by IRDye 800CW-conjugated anti-rabbit IgG secondary antibody (LI-COR Biosciences, Lincoln, NE, USA). The signal was visualized using the Odyssey CLx Imaging System (LI-COR Biosciences). Normal human plasma (Siemens Healthineers, Erlangen, Germany) was used as a VWF multimer control for each gel. Although we attempted to perform multimer analysis of the conditioned media collected on day six from shRNA-transfected cells, similar to the immunofluorescence (see above), we found that a considerable number of cells were detached from plates during 2 hr of rest. We believe that the depletion of the V0a isoform disturbs the adherence of HUVECs; thus, we used conditioned media collected from day 5 culture for the analysis (*Figure 3—figure supplement 1B*). We performed CBB staining of cell lysate to confirm equal loading of the samples in each experiment.

## Acknowledgements

Authors are grateful to Hiroko Shirotani-Ikejima and Taiko Kunieda for their technical assistance.

## Additional information

### Funding

| Funder | Grant reference number | Author |
|---|---|---|
| Japan Society for the Promotion of Science | 17K07401 | Yasuo Yamazaki |
| Japan Society for the Promotion of Science | 17K07358 | Koichi Kokame |
| Ministry of Health, Labour and Welfare | JPMH20FC1024 | Koichi Kokame |
| SENSHIN Medical Research Foundation | | Yasuo Yamazaki |
| Japan Agency for Medical Research and Development | JP20ek0210154 | Koichi Kokame |

The funders had no role in study design, data collection and interpretation, or the decision to submit the work for publication.

### Author contributions

Yasuo Yamazaki, Conceptualization, Data curation, Funding acquisition, Investigation, Project administration, Validation, Visualization, Writing - original draft; Yuka Eura, Investigation, Writing - review and editing; Koichi Kokame, Data curation, Funding acquisition, Project administration, Validation, Writing - review and editing

### Author ORCIDs

Yasuo Yamazaki (ID) http://orcid.org/0000-0001-5297-9837
Yuka Eura (ID) http://orcid.org/0000-0001-5784-6339
Koichi Kokame (ID) http://orcid.org/0000-0002-9654-6299

### Decision letter and Author response

Decision letter https://doi.org/10.7554/eLife.71526.sa1
Author response https://doi.org/10.7554/eLife.71526.sa2

## Additional files

### Supplementary files

• Transparent reporting form

### Data availability

All data generated or analysed during this study are included in the manuscript and supporting file; Source Data files have been provided for all blots and graphs shown in the manuscript.

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
