## [Editor Report]

This study is of particular interest to endothelial cell biologists and cell biologists working on intracellular transport. The experiments provide new insights into the contribution of a proton pump and protein kinase D in the formation of Weibel-Palade bodies that contain von Willebrand factor in vascular endothelial cells.

---

## [Decision Letter]

**Decision letter after peer review:**

Thank you for submitting your article "V-ATPase V0a1 promotes Weibel-Palade body biogenesis through protein kinase D-dependent membrane fission" for consideration by *eLife*. Your article has been reviewed by 3 peer reviewers, and the evaluation has been overseen by a Reviewing Editor and Suzanne Pfeffer as the Senior Editor. The following individuals involved in review of your submission have agreed to reveal their identity: Felix Campelo (Reviewer #1); Vivek Malhotra (Reviewer #2).

The reviewers have discussed their reviews with one another, and the Reviewing Editor has drafted this to help you prepare a revised submission. We include the full reviews to help guide your next steps in revising this manuscript.

Essential revisions:

The reviewer’s request:

1. Quantitative assessment of changes in WPB size and shape in V0a2/V0a1 depleted cells.

2. Measure lumenal Golgi/TGN pH in WT and a1/a2 KD cells.

3. The claim that PKD is active in the shV0a1 cells is not supported by the data. Please measure PKD activity using more adequate assays (phospho-specific proteins etc) in the absence of V0a1. it seems that PKD in V0a1-depleted cells is predominantly at the Golgi. Therefore, detection of PKD activity by WB using the commercially available activation-loop Ab or an IVK should reflect Golgi activity. Alternatively, you could also use a genetically encoded reporter to measure PKD activity.

4. Provide a more thorough discussion including how pH and lipid metabolism (DAG, PI4P) might be related.

*Reviewer #1:*

In this work, the authors used human umbilical vein endothelial cells (HUVEC cells) to investigate the biogenesis of Weibel-Palade bodies (WPBs). Since it is well known that an acidic pH in this compartment is crucial for the hemostatic function of its most important component – the protein von Willebrand factor (VWF) -, the authors investigated the role of a particular set of V-ATPase subunits in WPB biogenesis.

They reported that both the endogenous and the exogenously-expressed GFP-tagged V-ATPase subunits V0a1 and V0a2 localize to WPBs. Particularly, V0a1 appears to be found in peripheral (probably mature) WPBs, whereas V0a2 localized to perinuclear regions containing VWF (TGN/nascent WPBs). The authors then found that these proteins are required for the proper morphology of WPBs: when V0a2 is absent, WPBs formed at normal rates but present twisted shapes; whereas when V0a1 is absent, less WPBs are found in cells, and those that form are located at close apposition (maybe still connected to) of the TGN. The authors therefore suggested that V0a1 is a component required for the fission from the TGN of nascent WPBs.

Next, they investigated known TGN fission factors, and found the involvement of protein kinase D (PKD) in WPB formation. Specifically, when expressing kinase-dead (KD) mutants of PKD – which form long tubes connected to the TGN and enriched with cargo proteins (because of defective carrier fission) -, they found that VWF was included in those PKD-KD tubes, suggesting that VWF is a PKD-dependent cargo protein. It is not clear how PKD is required for WPB biogenesis, and what, if any, is the connection between V0a1 and PKD in this process. Interestingly, there's reports in the literature (e.g. doi:10.1152/ajpcell.00504.2008) that show that in the absence of PKD1, endothelial cells can still secrete normal levels of VWF. Finally, the authors suggested that PKDs present in nascent WPBs are enzymatically active because they reported that DAG and PI4P sensors are present on those membranes. However, normal levels of PI4P do not necessarily need to report an enzymatically active PKD, since, for instance, active PKD has been shown to reduce PI4P levels at TGN membranes (by phosphorylation of its substrate OSBP). Hence, it is not clear how V0a1 promotes WPB biogenesis in a PKD-dependent manner.

In summary, this article presents new data showing the involvement of V0a1 in WPB biogenesis, and reported the presence of VWF proteins in PKD-KD tubules. These results open new questions in the field, especially on the mechanism by which these two proteins control WPB fission, and if there is some interplay between them in doing so.

Major comments:

As I stated above, I think the results are potentially interesting, but it is not clear to me what is the molecular mechanism by which V0a1 controls WPB biogenesis. The link with PKD is mild (as said earlier, there are reports indicating that PKD1 is dispensable for VWF secretion, which need to be cited and put in context of the present findings). Some ideas:

– V-ATPases regulate luminel pH. In the absence of V0a1 (or V0a2), is there any changes in WPBs pH? The authors reported that VWF still form multimers, but this only quite indirectly reports on the pH at the TGN/WPBs.

– About the involvement of PKDs: for the quantitation shown in Figure 5A (and related supps), it would help to count the number of WPBs in the cells expressing the PKD-KD mutants, similarly to what has been done in Figure 4B for the knockdowns of the V0as. And as said, put that in the context of the existing literature on the topic.

– Line 240-41: according to the data, both V0a1 knockdown and PKD-KD expression show a similar phenotype (fission defect). What is the connection (if any) between these two processes?

– Figure 6: PKD activity is assessed by the presence of PI4P on the TGN-associated VWF-positive buds. However, PKD is also phosphorylating OSBP, which reduces PI4P levels (actually, acute activation of PKD leads to less PI4P in the Golgi). There are other means to establish PKD activity (use of phospho-substrate antibodies, etc.)

– Lack of mechanisms or upstream/downstream organisation of the V0as/PKD in fission. Do V0a localize to PKD-KD tubes? Is V0a1 a substrate of PKD?

*Reviewer #2:*

This is a nice story that merits publication because it addresses a long standing issue on the export of bulky WPBs from the Golgi. The involvement of VO-ATPase subunits and PKD pathway makes this process amenable to molecular analysis. My gut feeling is that trafficking of WPBs is likely related to the export of other bulky cargoes like the collagens.

The migration or trafficking of WPB's remains a fascinating challenge. The data on different location of VOa1 and VOa2 are very interesting. The question I have is whether there are different pools of VO-ATPases? Can the authors isolate ( immuno precipitate) two different pools of VO-ATPase; one enriched in VOa1 and another in VOa2? Or did I get this wrong and these subunits are not in complex with other parts of VO-ATPase.

The data on the PKD-DAG are very interesting, but I am a bit confused about the involvement of clathrin. My understanding is that DAG-PKD pathway is independent of the clathrin mediated cargo export from the TGN. Is clathrin found on PKD-Kinase dead containing tubules that are enriched in WBPs?

Finally, is this a budding event or is it that PKD-DAG sequester a part of TGN enriched in WBPs , while the unoccupied part concentrates cargoes exported by the clathrin pathway? The authors should explain this. It would be terrific to test whether a cargo destined to the endosomes/lyososomes is segregated from WPBs in the TGN.

*Reviewer #3:*

The formation of Weibel-Palade bodies (WPBs), the secretory granules of endothelial cells, is driven by von Willebrand factor (VWF), their most abundant cargo protein. VWF is a large multidomain and multimeric protein that further multimerises in the Golgi complex after dimerisation in the ER, leading to the formation of VWF tubules that are packed into nascent WPBs in the TGN. The number and size of WBPs formed is ultimately controlled by the number of VWF quanta (VWF dimers fused together) formed in the Golgi complex. Importantly, the acidic luminal environment of the Golgi complex and transport vesicles is required for conformational rearrangements of VWF dimers, their tubulation and packing of multimers, respectively, and thus determines the WPB size. Since WPB size determines endothelial function, it is important to understand the mechanism underlying VWF tubulation and multimerisation. The multivacuolar (V)-ATPase-mediated proton pump consists of two large domains, V0 and V1, and continuously pumps H^+^ ions into the Golgi lumen, contributing to the acidic pH of the organelle. While the contribution of V-ATPase activity to membrane trafficking is known, the V0a isoform involved in the formation of WPBs at the TGN has not yet been identified.

In this paper, the authors have identified two V0 isoforms, V0a1 and V0a2, to be associated with mature WPBs and nascent WPBs, respectively. By using confocal microscopy and biochemical analysis, they show that the two isoforms do not contribute to multimerisation of VWF but are instead required for biogenesis of WPBs at the TGN. Furthermore, they propose that downstream of V0a1 protein kinase D (PKD) is required for WPB membrane fission at the TGN. The paper thus sheds light on a potential function of V-ATPase V0a1 in WPB formation and adds important information on isoform-specific functions of this member of the V-ATPase family. However, while the data showing the localization to and involvement of V0a1 in the biogenesis of WPBs are convincing, the experiments on the molecular mechanisms, especially the link to PKD-dependent membrane fission, fall short and are insufficient to support the conclusions.

Figure 1, Supplement 3F: V0a4-EGFP shows considerable co-localization with peripheral VWF-positive structures, however, the authors do not pursue this finding further claiming that V0a1 and V0a2 are the pre-dominant isoforms located on WPBs. The authors should have investigated a potential effect of V0a4 to rule out that this isoform is involved in WPB biogenesis/transport.

Figure 4: The finding that V0a1 is found on mature WPBs but its depletion affects WPB formation at the TGN is exciting. To make their data more solid the authors should have investigated if and how the pH in the Golgi/TGN changes upon depletion of V0a1. For example, they could have used a pH biosensor.

Figure 5: The authors find that PKD family is involved in the biogenesis of WPBs. While these data are convincing the link to V0a1 is insufficient. Upon V0a1 knockdown more PKD is found on the TGN, which might indicate that more DAG is present. This is an interesting finding, however, not further pursued by the authors. The authors use several PI4P biosensors to show that PKD is enzymatically active. It is clear that there is increased binding of these biosensors to the TGN when V0a1 is depleted, however, many pathways besides PKD contribute to PI4P production and therefore additional evidence should have been provided.

Additionally, PI4P has been shown to sense the cytosolic pH and this impacts cargo sorting and trafficking. Acidification of cytosolic pH displaces PH domains from PI4P (Shin et al., Dev Cell, 2020). Given these results, the authors should have investigated whether silencing of V0a1 could also affect intracellular pH and thus the binding of PH domains to PI4P.

General comment: The authors make several references to morphological changes in WPB size and shape upon perturbation (e.g. Figure 3C). However, it is difficult to see these changes when looking at the images. In particular, in Figure 3A, the difference in VWF-positive structures between shControl and shV0a2 is not clear. Given the findings that number and size of WPBs might be affected by V0a1 and V0a2 depletion but mulitmerization and secretion are not impacted or even increased (Figure 3 – supplement 1) the authors should have considered to investigate VWF string formation as this is dependent on size of WPBs and VWF tubulation.

Figure 1: The authors use Bafilomycin A1, a specific vacuolar H^+^ ATPase (V-ATPase) inhibitor, to investigate the contribution of V-ATPase to WPB formation. They show that BafA1 treatment does not affect the amount basal and regulated VWF secretion in general but disrupts VWF multimerization. While the data shown are convincing they are in conflict with data published by Torisu et al., (2013) who showed that inhibition of autophagy by BafA1 treatment reduces the secretion of VWF from endothelial cells. Some discussion would be needed to clarify the discrepancy.

Figure 5: The authors could use published PKD activity reporters, perform an in vitro kinase assay, or measure activation loop phosphorylation of PKD via Western Blot analysis to infer on PKD activity in V0a1 depleted cells. Also, some discussion on how V0a1 depletion might affect DAG levels at the TGN would be needed.

General:

The authors should assess changes in WPB size and morphology using quantitative morphometric analysis of the VWF-positive structures (see Lopes da Silva et al., JCellSci, 2016).

[Editors’ note: further revisions were suggested prior to acceptance, as described below.]

Thank you for resubmitting your work entitled "V-ATPase V0a1 promotes Weibel-Palade body biogenesis through protein kinase D-dependent membrane fission" for further consideration by *eLife*. Your revised article has been evaluated by Suzanne Pfeffer (Senior Editor) and a Reviewing Editor.

The manuscript has been improved but there are some remaining issues that need to be addressed, as outlined below:

– Provide some more complete quantitation of WPBs shape. For that they can measure the two Feret's diameters (short and long) for each granule, and from there the aspect ratio (in this case, higher aspect ratio means straighter WPB, lower aspect ratio means more twisted).

– Revise the text to be careful when claiming a specific role of PKD in this process.

– provide a rescue experiment on the effect of V0a1 depletion on PKD1 expression levels because the authors do not use a second, independent shRNA for V0a1 depletion.

*Reviewer #1:*

The authors have revised the manuscript and presented a detailed response to the reviewers' questions and comments. From my side, although I very much value the new data provided by the authors, I am still not fully convinced that all the conclusions that the authors claim in this manuscript are well sustained by the data. In particular, in terms of the statement that V0a1 promotes WPB formation through a PKD-dependent membrane fission. I am still quite positive on this manuscript, but I still have the following comments regarding the new data provided by the authors:

– Main point #1: While I do appreciate that the authors have presented quantitative information of WPB size in their revised manuscript, I'm not fully convinced that presenting a only the measurement of the large Feret's is the best way to quantitate shape of the WPB. E.g. in pg 9 they say that for a twisted WPB, one would expect a shorter Feret's diameter, which is true if the WPB have the same size, but it is not a direct measure of "twist" (different Feret's could be caused by other shape changes as compared to twisting). Also, the authors do not present the total distribution of Feret's but otherwise the rather arbitrary % of WPBs larger than 2um. I think that the phentype based on the few images provided is clear, but I think a better quantitative analysis could have been performed.

– Main point #3: Pg. 10 the authors say "In the absence of V0a1, WPBs 238 were unable to separate from the TGN, although the buds were formed (Figure 4).". I still am not convinced that this claim is sustained by the experimental data. Also, the new data on PKD1 overexpression is quite interesting. What about PKD2 and 3, have't the authors tested that? Is the high PKD1 expression an off.target effect (could the authors test it in the rescue experiments)? Also I'm puzzled on why the authors used an anti-PKD2 antibody in Figure 5supp4 and not an anti-PKD1, since they shown that PKD1 is upregulated in shV0a1 cells?

*Reviewer #3:*

The authors have put a lot of effort into revising the manuscript and have addressed the major criticisms. Particularly valuable in this regard are the experiments on intraluminal pH in the Golgi that support the hypothesis that WPB formation occurs at specific subdomains of the TGN. The experiments also revealed that PKD activity is not dependend on V0a1 and therefore the relationship between V0a1 and PKD activity remains unresolved. However, in my opinion, this does not prevent publication of the manuscript.

---

## [Author Response]

Essential revisions:The reviewer’s request:1. Quantitative assessment of changes in WPB size and shape in V0a2/V0a1 depleted cells.

We thank the reviewers for this comment. To clarify the difference of WPB size and shape generated in the absence of V0a subunits, we have measured Feret’s diameter, the longest distance between any two points along individual WPB particles. Feret’s diameter is expected to be smaller if WPBs are twisted or in irregular shapes compared with relatively straight structures observed in the control. The data is shown as a percentage of WPBs of which Feret’s diameter is larger than 2 µm long (Figure 4B right). As expected from the confocal images, Feret’s diameter of WPBs in the V0a2-depleted cells was significantly smaller than those in the control. We also found that Feret’s diameter of WPBs in the V0a1-depleted cells were even smaller than those in the V0a2-depleted cells. We think that this represents the appearance of the rounded WPBs found in cell periphery in the V0a1-depleted cells (Figure 4A), although the WPBs accumulated around the TGN still retain the characteristic elongated shape. As the rounded WPBs were not that much observed when WPBs were visualized by the introduction of VWF^D1-A1^-EGFP and P-selectin-EGFP in the V0a1-depletion confirmed cells (Figure 4D,E), we concluded that they are WPBs formed before the V0a1 deletion is achieved. In regards to this, we have amend the text (Lines 195 to 199) and the corresponding figure legends (Lines 691 to 701).

2. Measure lumenal Golgi/TGN pH in WT and a1/a2 KD cells.

To respond to this comment, we have added new data introducing β-1,4-galactosyltransferase 1 (GalT), a well-characterized luminal TGN resident protein coupled with pHluorin, a pH-sensitive GFP derivative (GalT-pHluorin). This data has been included as Figure 3—figure supplement 2 in the revised manuscript. Bafilomycin A1 and concanamycin A exposure caused almost total neutralization of the luminal TGN pH, while the depletion of the V0a isoforms resulted in modest effects. These observations are consistent with our data showing that V-ATPase inhibition resulted in the blockage of VWF multimerization, but the depletion of V0a isoforms did not (Figure 2—figure supplement 1A, Figure 3—figure supplement 1B), because the multimerization requires acidic pH. The V0a2-depletion resulted in a slight but homogenous pH increase over the TGN (Figure 3—figure supplement 2B). In contrast, we have found relatively high pH increase in some, but not all of the TGN subdomains in the V0a1-deplted cells (Figure 3—figure supplement 2B), although no statistical difference was observed when the whole TGN pH was compared (Figure 3—figure supplement 2A). We think the impacts of V0a1-depletion in the pH regulation of particular TGN subdomains is interesting, because the main cellular pool was found in peripheral WPBs (Figure 1). WPBs were found to form a close contact with endosomes by electron tomography analyses. We think that this suggest a way how V0a1 is delivered to WPBs during the biogenesis, and the specific effects on some TGN subdomains suggest that WPB biogenesis occurs in particular subdomain of the TGN. In regards to the above results and discussion, we have amended the text in the results and Discussion sections (Lines 171 to 186, 320 to 322) as well as in the corresponding figure legend (Lines 844 to 853) and the Materials and methods section (Lines 446 to 452).

3. The claim that PKD is active in the shV0a1 cells is not supported by the data. Please measure PKD activity using more adequate assays (phospho-specific proteins etc) in the absence of V0a1. it seems that PKD in V0a1-depleted cells is predominantly at the Golgi. Therefore, detection of PKD activity by WB using the commercially available activation-loop Ab or an IVK should reflect Golgi activity. Alternatively, you could also use a genetically encoded reporter to measure PKD activity.

Thank you very much for the comments. As the reviewers pointed, there are several routes to supply PI4P to the Golgi/TGN other than the PKD-mediating pathway. So we agree that PKD activity should be evaluated by a more relevant assay in addition to the data using PI4P reporters (Figure 6C, Figure 6—figure supplement 1B,C). To see the PKD activity in the absence of V0a1, we performed western blotting using anti-phospho-PKD1(Ser744/748) and anti-total PKD1 antibodies (Cell Signaling Technology) as the reviewers suggested. These data have been included as Figure 6–supplement 2 in the revised manuscript. The depletion of V0a1 unexpectedly caused an increase of total PKD1 protein, whereas we were unable to see any detectable changes of the phospho-PKD1 level compared to the control. As the substantial amount of PKDs was observed on the nascent WPB in the absence of V0a1 compared with the cytosolic distribution in the control (Figure 5B-E, Figure 5—figure supplement 3D,E), an increase of PKD1 protein in the V0a1-depleted cells may be explained by a difference in the protein stability dependent on the subcellular localization. Alternatively, it might be an upregulation of PKD1 expression by a cell autonomous compensation or unknown mechanism mediated by the V0a1-depletion. Because of the low endogenous expression of PKDs, it is hard to test whether the PKDs observed on the accumulated WPBs in the V0a1-depleted cells are active/phosphorylated or not. Therefore, although total amount of cellular active PKD1 is unchanged independent from the absence or presence of V0a1, we were unable to clarify the requirement of V0a1 on the PKDs’ activation/phosphorylation. Regarding the above results and discussion, we have included appropriate text in the result section (Lines 275 to 297) and the corresponding figure legend (Lines 914 to 918).

4. Provide a more thorough discussion including how pH and lipid metabolism (DAG, PI4P) might be related.

As discussed above, V0a1 appears to be involved in the regulation of TGN pH. Although it is unclear yet if V0a1-mediating pH change triggers membrane fission of WPBs, we think that pH-dependent biogenesis of membrane-bound structures is feasible. We have shown that PI4P and DAG are observed in newly forming WPB buds; it is particularly prominent in the V0a1-depleted cells (Figure 6, Figure 6—figure supplement 1). As these lipids were not found in peripheral WPBs, they must be removed or converted into other lipids during the biogenesis, suggesting a dynamic lipid metabolism in WPB membrane. PI4P and phosphatidic acid, a lipid generated through single-step metabolism from DAG, are shown to work as pH biosensors in cells, and the pH-dependent interaction with particular proteins regulates cellular processes such as biogenesis of membrane-bound structures and intracellular trafficking (Shin *et al., Dev Cell* 2020, Young *et al., Science* 2010). Thus, we think that possible involvement of lipids and TGN luminal pH in the biogenesis of WPBs should be tested in the future. Regarding this, we have added a discussion (Lines 338 to 345).

Reviewer #1:Major comments:As I stated above, I think the results are potentially interesting, but it is not clear to me what is the molecular mechanism by which V0a1 controls WPB biogenesis. The link with PKD is mild (as said earlier, there are reports indicating that PKD1 is dispensable for VWF secretion, which need to be cited and put in context of the present findings). Some ideas:– V-ATPases regulate luminel pH. In the absence of V0a1 (or V0a2), is there any changes in WPBs pH? The authors reported that VWF still form multimers, but this only quite indirectly reports on the pH at the TGN/WPBs.

Thank you very much for your comments. We have measured intraluminal pH of the TGN where WPBs are generated. Please refer our response to Essential revision #2.

– About the involvement of PKDs: for the quantitation shown in Figure 5A (and related supps), it would help to count the number of WPBs in the cells expressing the PKD-KD mutants, similarly to what has been done in Figure 4B for the knockdowns of the V0as. And as said, put that in the context of the existing literature on the topic.

We have added the WPB counts of PKD-KD expressing cells as Figure 5—figure supplement 3C in the revised manuscript. Thank you very much for the pointing the paper that was missed in the original manuscript. It has been cited in the discussion in the revised manuscript (Line 333-334, Reference #74). As we observed essentially same effects in all PKDs, we consider that PKDs are redundant each other in the context of WPB biogenesis as described in the discussion (Lines 327 to 334).

– Line 240-41: according to the data, both V0a1 knockdown and PKD-KD expression show a similar phenotype (fission defect). What is the connection (if any) between these two processes?

Thank you very much for your comments. As the fission defect phenotype observed in the absence of V0a1 was not rescued even by the induction of wild-type PKDs (Figure 5B-E, Figure 5—figure supplement 3D,E), we believe that V0a1 must be a primary factor required for the fission, although the connection between V0a1 and PKDs is uncertain yet. We would like to address on this in our future study.

– Figure 6: PKD activity is assessed by the presence of PI4P on the TGN-associated VWF-positive buds. However, PKD is also phosphorylating OSBP, which reduces PI4P levels (actually, acute activation of PKD leads to less PI4P in the Golgi). There are other means to establish PKD activity (use of phospho-substrate antibodies, etc.)

Please refer our response to Essential Revision #3.

– Lack of mechanisms or upstream/downstream organisation of the V0as/PKD in fission. Do V0a localize to PKD-KD tubes? Is V0a1 a substrate of PKD?

We attempted to see if endogenous V0a1 protein is localized to PKD-KD tubes; however, we were unable to obtain adequate resolution data that allows us to draw a conclusion, because of the broad punctate distribution of endogenous V0a1 protein in cells.

Reviewer #2:This is a nice story that merits publication because it addresses a long standing issue on the export of bulky WPBs from the Golgi. The involvement of VO-ATPase subunits and PKD pathway makes this process amenable to molecular analysis. My gut feeling is that trafficking of WPBs is likely related to the export of other bulky cargoes like the collagens.The migration or trafficking of WPB's remains a fascinating challenge. The data on different location of VOa1 and VOa2 are very interesting. The question I have is whether there are different pools of VO-ATPases? Can the authors isolate ( immuno precipitate) two different pools of VO-ATPase; one enriched in VOa1 and another in VOa2? Or did I get this wrong and these subunits are not in complex with other parts of VO-ATPase.

Thank you very much for your comments. V0 and V1 sectors of V-ATPase are shown to associate reversibly dependent on cellular condition such as nutrient status; suggesting that V0a isoforms found in cells are not always in "complete" V-ATPase complex. As shown in Figure 1—figure supplement 3A,B, we found that V1c and V1A, common subunits in the V0 and the V1 sectors, are predominantly observed in WPBs in HUVECs. Thus, it is likely that V0a isoforms found on WPBs are in complete V-ATPase complex. Thus, we believe that there are two distinct pools of V-ATPase complex in HUVECs; however, because of the lack of specific tools/antibodies, currently it is hard for us to test the hypothesis.

The data on the PKD-DAG are very interesting, but I am a bit confused about the involvement of clathrin. My understanding is that DAG-PKD pathway is independent of the clathrin mediated cargo export from the TGN. Is clathrin found on PKD-Kinase dead containing tubules that are enriched in WBPs?

As PKDs are shown to be involved in transport carrier biogenesis of various cargos, we believe that careful approaches are required to respond to the comments. We would like to address on this in our future study. Thank you very much for your constructive comments.

Finally, is this a budding event or is it that PKD-DAG sequester a part of TGN enriched in WBPs , while the unoccupied part concentrates cargoes exported by the clathrin pathway? The authors should explain this. It would be terrific to test whether a cargo destined to the endosomes/lyososomes is segregated from WPBs in the TGN.

Thank you very much for your comments. We also believe that cargos destinated to endosomes/lysosomes are segregated from WPBs in the TGN. In the revised manuscript, we have included a data showing that V0a1 appears to be involved in the pH regulation of some TGN subdomains. We think that it suggests WPB formation in particular subdomains of the TGN. We would like to address on this carefully together with #2-2 in our future study.

Reviewer #3:Figure 1, Supplement 3F: V0a4-EGFP shows considerable co-localization with peripheral VWF-positive structures, however, the authors do not pursue this finding further claiming that V0a1 and V0a2 are the pre-dominant isoforms located on WPBs. The authors should have investigated a potential effect of V0a4 to rule out that this isoform is involved in WPB biogenesis/transport.

Thank you very much for your comments. As no suitable antibodies to V0a4 were available, we were unable to confirm whether endogenous V0a4 protein is localized on WPBs. Because of the specific phenotypes upon V0a1/V0a2-depeltion, we believe that V0a4 must have a distinct role other than the roles described in this study.

Figure 4: The finding that V0a1 is found on mature WPBs but its depletion affects WPB formation at the TGN is exciting. To make their data more solid the authors should have investigated if and how the pH in the Golgi/TGN changes upon depletion of V0a1. For example, they could have used a pH biosensor.

Please refer our response to Essential Revision #2.

Figure 5: The authors find that PKD family is involved in the biogenesis of WPBs. While these data are convincing the link to V0a1 is insufficient. Upon V0a1 knockdown more PKD is found on the TGN, which might indicate that more DAG is present. This is an interesting finding, however, not further pursued by the authors. The authors use several PI4P biosensors to show that PKD is enzymatically active. It is clear that there is increased binding of these biosensors to the TGN when V0a1 is depleted, however, many pathways besides PKD contribute to PI4P production and therefore additional evidence should have been provided.Additionally, PI4P has been shown to sense the cytosolic pH and this impacts cargo sorting and trafficking. Acidification of cytosolic pH displaces PH domains from PI4P (Shin et al., Dev Cell, 2020). Given these results, the authors should have investigated whether silencing of V0a1 could also affect intracellular pH and thus the binding of PH domains to PI4P.

Please refer our responses to Essential Revision #2 and #3.

General comment: The authors make several references to morphological changes in WPB size and shape upon perturbation (e.g. Figure 3C). However, it is difficult to see these changes when looking at the images. In particular, in Figure 3A, the difference in VWF-positive structures between shControl and shV0a2 is not clear. Given the findings that number and size of WPBs might be affected by V0a1 and V0a2 depletion but mulitmerization and secretion are not impacted or even increased (Figure 3 – supplement 1) the authors should have considered to investigate VWF string formation as this is dependent on size of WPBs and VWF tubulation.

Regarding the WPB size and shape in the absence of V0a2, please refer our response to Essential Revision #1. We did not perform any analysis of the regulated secretion of VWF including VWF string formation using shRNA-introduced cells, because we think that use of those cells makes data interpretation difficult. As discussed in the manuscript (Lines 204 to 208), WPBs are storage organelles; that is, once formed, they remain in cell until they are being exocytosed or turned over. Efficient reduction of target proteins by shRNAs requires time. Thus, it is likely that shRNA-introduced cells contain not only the WPBs that are formed in the absence of the target protein, but also the pre-formed WPBs before the depletion is achieved. Therefore, the regulated secretion from shRNA-introduced cells must include sum of VWF from heterogenous populations of WPBs. And also, there is a possibility that their response to exocytic stimulants might be different. From the above reasons, we do not think we can find anything relevant from the experiments using shRNA-introduced cells in terms of the regulated secretion.

Figure 1: The authors use Bafilomycin A1, a specific vacuolar H^+^ ATPase (V-ATPase) inhibitor, to investigate the contribution of V-ATPase to WPB formation. They show that BafA1 treatment does not affect the amount basal and regulated VWF secretion in general but disrupts VWF multimerization. While the data shown are convincing they are in conflict with data published by Torisu et al., (2013) who showed that inhibition of autophagy by BafA1 treatment reduces the secretion of VWF from endothelial cells. Some discussion would be needed to clarify the discrepancy.

Thank you very much for your comments. To see the effects of V-ATPase blockage, we used HUVECs exposed to the V-ATPase inhibitors for up to 4 hours, because we found that relatively short exposure was enough to see the effects and that the prolonged exposure (16 hours) induces a formation of large vesicles/vacuoles that can be observed even under phase-contrast microscopy. In the paper, they used HUVECs exposed to bafilomycin A1 for 16 hours. So we think that the discrepancy is due to the difference in the conditions. In regards to this, we have added a note in the legend of Figure 1—figure supplement 2B (Lines 774-775).

Figure 5: The authors could use published PKD activity reporters, perform an in vitro kinase assay, or measure activation loop phosphorylation of PKD via Western Blot analysis to infer on PKD activity in V0a1 depleted cells. Also, some discussion on how V0a1 depletion might affect DAG levels at the TGN would be needed.

Please refer our responses to Essential Revision #3 and #4.

General:The authors should assess changes in WPB size and morphology using quantitative morphometric analysis of the VWF-positive structures (see Lopes da Silva et al., JCellSci, 2016).

Please refer our response to Essential Revision #1.

[Editors' note: further revisions were suggested prior to acceptance, as described below.]

The manuscript has been improved but there are some remaining issues that need to be addressed, as outlined below:– Provide some more complete quantitation of WPBs shape. For that they can measure the two Feret's diameters (short and long) for each granule, and from there the aspect ratio (in this case, higher aspect ratio means straighter WPB, lower aspect ratio means more twisted).

Thank you very much for the comments. We have measured the longest (F_max_) and shortest (F_min_) Feret’s diameters of individual WPB particles to calculate F_max_/F_min_ ratio as suggested. Frequency distribution of the ratio (F_max_/F_min_) as well as the longest Feret’s diameter (F_max_) has been shown as histograms in Figure 3—figure supplement 3 to present total WPB distribution. Obvious decrease in F_max_/F_min_ is observed in WPBs formed in the absence of V0a2; indicating the irregular, twisted shape as seen in the images (Figure 3). Regarding this, we have amended text in the result section (Lines 201-206) and the corresponding figure legend (Lines 880-889). And we have removed a figure panel in Figure 4B showing a comparison of WPB proportions larger than 2 µm and the corresponding text (Lines 706-714).

– Revise the text to be careful when claiming a specific role of PKD in this process.

Thank you very much for the comments. Because we observed that the induction of wild-type PKDs does not support the segregation of WPBs in the absence of V0a1 (Figure 5B, Figure 5—figure supplement 3D,E), we think it indicates a primary role of V0a1 in the membrane fission of WPBs, although the molecular link between V0a1 and PKD in WPB biogenesis is still unclear. To respond to this comment, we have carefully amended the text throughout the manuscript including the title (Lines 2-3, 29-32, 275-276, 316, 362-366, 728-729, and 765-771).

– provide a rescue experiment on the effect of V0a1 depletion on PKD1 expression levels because the authors do not use a second, independent shRNA for V0a1 depletion.

Thank you very much for the comments. To respond to this, we have included new data in Figure 6—figure supplement 2B showing that a second, independent shRNA against V0a1 also caused an increase of PKD1 protein as similarly observed in the other shRNA. We also performed a rescue experiment by the induction of HA-Va01 as suggested, however, we were unable to observe any rescue effects on the PKD1 protein level. Because we observed the obvious rescue effects in the fission defect phenotype (Figure 4C), we believe HA-V0a1 must be functional in this context. We assume that HA-V0a1 might not be functional enough regarding the regulatory function of PKD1 protein level or alternatively the overexpression may cause something unexpected. In conclusion, as two independent shRNAs targeting distinct region of V0a1 mRNA (UTR and Tyr^620^-Met^627^) resulted in the similar induction, it is likely that V0a1 depletion itself causes an induction of PKD1 protein in HUVECs.

Reviewer #1:The authors have revised the manuscript and presented a detailed response to the reviewers' questions and comments. From my side, although I very much value the new data provided by the authors, I am still not fully convinced that all the conclusions that the authors claim in this manuscript are well sustained by the data. In particular, in terms of the statement that V0a1 promotes WPB formation through a PKD-dependent membrane fission. I am still quite positive on this manuscript, but I still have the following comments regarding the new data provided by the authors:

Thank you very much for your comments. Please refer our response to the comment above.

– Main point #1: While I do appreciate that the authors have presented quantitative information of WPB size in their revised manuscript, I'm not fully convinced that presenting a only the measurement of the large Feret's is the best way to quantitate shape of the WPB. E.g. in pg 9 they say that for a twisted WPB, one would expect a shorter Feret's diameter, which is true if the WPB have the same size, but it is not a direct measure of "twist" (different Feret's could be caused by other shape changes as compared to twisting). Also, the authors do not present the total distribution of Feret's but otherwise the rather arbitrary % of WPBs larger than 2um. I think that the phentype based on the few images provided is clear, but I think a better quantitative analysis could have been performed.

Thank you very much for your suggestion. Please refer our response to the comment above.

– Main point #3: Pg. 10 the authors say "In the absence of V0a1, WPBs 238 were unable to separate from the TGN, although the buds were formed (Figure 4).". I still am not convinced that this claim is sustained by the experimental data. Also, the new data on PKD1 overexpression is quite interesting. What about PKD2 and 3, have't the authors tested that? Is the high PKD1 expression an off.target effect (could the authors test it in the rescue experiments)? Also I'm puzzled on why the authors used an anti-PKD2 antibody in Figure 5supp4 and not an anti-PKD1, since they shown that PKD1 is upregulated in shV0a1 cells?

Thank you very much for your comments. Regarding the sentence "In the absence of V0a1, WPBs were unable to separate from the TGN, although the buds were formed (Figure 4)", we have amended to " In the absence of V0a1, WPBs seemed unable to separate from the TGN, although the buds were formed (Figure 4)". Regarding the concern of off target effect of shRNA, please refer our response to the comment #C. As no suitable immunofluorescence antibodies to PKD1 were available, we were unable to confirm where endogenous PKD1 protein is localized in the absence of V0a1. Because we think that our results suggest a functional redundancy of PKDs in this process, we looked at PKD2 instead.